# Environmentally adjusted $\delta^{13}$C thresholds for accurate detection of C$_4$ plant consumption in Europe
Margaux L. C. Depaermentier [1,3] ✉, Michael Kempf [2,3] ✉ & Giedrė Motuzaitė Matuzevičiūtė [1]

Detecting C$_4$ plants consumption is central to investigating animal ecology, agriculture, dietary transitions, and socio-environmental adaptations, and can be done using carbon isotope analysis. The conventional $\delta^{13}$C threshold used to identify C$_4$ plant intake does not consider substantial ecological variability across Europe. By analyzing over 4,000 $\delta^{13}$C values from archaeological C$_3$ and C$_4$ grains, we present a European-wide C$_3$ grain $\delta^{13}$C baseline and establish adjusted $\delta^{13}$C threshold estimations for C$_4$ consumption from the site to the ecozone scale using multicomponent environmental models and ecozone cluster analysis. We show that a fixed threshold lead to under- or overestimation of C$_4$ plant consumption, particularly in northern/humid and southern/arid regions, where the threshold needs to be revised downwards or upwards by up to 2‰. This refined framework offers a more accurate baseline for interpreting human and animal diet and enhances our understanding of the spread, adoption and consumption of C$_4$ crops across Europe.

Estimating the proportion of C$_3$ versus C$_4$ plants in human and animal diet is a key part of bioarchaeological, palaeontological and ecological research. Scholars worldwide have been investigating the spread of millet across Eurasia because this highly nutritious and drought-resistant C$_4$ plant can address various questions about past societies[1,2]. This includes complex social structures, the adoption of new subsistence strategies, the adaptation to challenging climatic and environmental settings, as well as mobility and health status[3–7]. In ecology and palaeontology, identifying C$_3$ and C$_4$ diets provides insight into (past) habitats, niche partitioning, and animal behaviours[8–12].

Stable carbon (C) isotope analyses—paired with nitrogen (N) isotope analyses when investigating collagen—represent the most efficient and preferred method to identify C$_4$ plant consumption from bioarchaeological and palaeontological skeletal remains. Due to different photosynthetic pathways between C$_3$ and C$_4$ plants, the ratio of $^{13}$C/$^{12}$C isotopes (expressed as $\delta^{13}$C in ‰) is significantly more negative in C$_3$ plants ($-35$ to $-23$‰) compared to C$_4$ plants ($-14$ to $-10$‰)[13–15]. Plant $\delta^{13}$C is transferred to the consumer's body tissues along the food chain, following well-described and quantified fractionation processes[16,17], which leads to enriched $\delta^{13}$C values in the tissues of C$_4$ plant consumers compared to C$_3$ plant consumers. In general, bone or dentine collagen $\delta^{13}$C values above $-18$‰ and enamel or bone apatite $\delta^{13}$C values above $-10$‰ indicate a mixed diet of C$_3$ and C$_4$ plants[13,18,19]. In contrast, $\delta^{13}$C values above $-12$‰ and $-4$‰, respectively, represent a pure C$_4$ diet[13,18,19].

However, specific environmental and climatic settings influence the plant's $\delta^{13}$C value[20]. For instance, C$_3$ plant $\delta^{13}$C values are more depleted under oceanic or Mediterranean climates, forest soil, dense canopy, elevated humidity or increased CO$_2$ concentration[20,21]. Conversely, continental climate, aridity, salinity (including sea-spray effect), elevated temperature or high altitude tend to enrich plant $\delta^{13}$C values, which are the basis of the terrestrial food chain[20,21]. The geographical location of the investigated site is also determinant. On average, skeletal tissues can be 1 to 2‰ lower in high latitudes compared to low latitudes in Europe[21–23]. Although C$_4$ plants react differently to environmental and climatic factors[14,24], their $\delta^{13}$C values can vary across latitude as well[25,26]. This implies that the threshold value for C$_4$ plant consumption needs to be adapted depending on the geographical location and environmental settings to avoid misinterpretations of body tissue isotopic composition. This paper fills this research gap to avoid an over- or underestimation of C$_4$ plants dietary intakes of premodern animals and human communities across Europe.

The main research questions addressed here are: (i) In which regions of Europe is it required to apply an environmentally adjusted $\delta^{13}$C threshold value for identifying C$_3$ versus C$_4$ plant consumption? (ii) What is the magnitude of this adjustment? (iii) Can we identify specific biogeographic parameters related to this isotope variability in Europe? This study draws on over 4000[27] published $\delta^{13}$C values from charred archaeological C$_3$ and C$_4$ grains derived from Isotope-Ratio Mass Spectrometry (IRMS)[28–101]. We

[1]Faculty of History, Vilnius University, Vilnius, Lithuania. [2]Department of Environmental Sciences, University of Basel, Basel, Switzerland. [3]These authors contributed equally: Margaux L. C. Depaermentier, Michael Kempf. ✉e-mail: margaux.depaermentier@if.vu.lt; michael.kempf@unibas.ch

present an innovative and, to the best of our knowledge, unprecedented ecozone-based model framework that integrates multivariate environmental data to facilitate the identification of $C_3$ versus $C_4$ plant diets in bioarchaeological, palaeontological and ecological research.

## Results

### Ecozone cluster model

Using multicomponent environmental datasets from topographical and climatic variables, we applied *k*-means cluster analysis to determine zones of similar environmental conditions across Europe. The model provides 20 spatial clusters including one cluster with unclassified values where not all conditions were equally met (Figs. 1 and S1, Table 1). Due to numeric quantization of *k* observations, we labelled the clusters based on the percentiles of the respective data ranges using expressional combinations of temperature, climatic moisture index (CMI), and topography. Some of them are geographically restricted to specific regions, such as the cold and humid ecozones 4 and 5 in North-Eastern Europe, the mild and very humid ecozone 8 at the Atlantic coast, or the hot and arid ecozones in the south of Europe, northern Africa and the Near East (e.g., clusters 10, 16 and 18). Other ecozones are spatially more scattered, for example European high mountain ranges (e.g., clusters 11 and 15). The results can be compared to

the biome-based ecoregions from the literature[102] despite the reduced number of modelled ecozones.

The sites from which the investigated grains originate cover 15 out of 20 modelled ecozones (Fig. 1, Supplementary data 1), representing most of the European geographical and ecological diversity. However, their distribution is biased by past and modern human activity, archaeological sampling and analytical strategies. Not all ecozones are equally represented in the sample and the ecozones 3, 12, 14, 15 and 18 were excluded from the analyses due to small sample sizes.

### Isotope diversity among grain species

Despite the isotopic diversity between the main $C_3$ grain species included in this study (Fig. S2A, B, Supplementary data 2), the two dominant crops of this sample, i.e. the *Triticum* (wheat, n = 1923) and *Hordeum* (barley, n = 1843) species, are largely overlapping in the northern and southern parts of Europe (Fig. S2B, Supplementary data 2). Differences from the Central/Western European samples are mostly caused by the wide geographical area represented by this region. In the UK, however, the two crops show notably distinct $\delta^{13}C$ values (Fig. S2B, Supplementary data 2), which possibly reflects species-specific differences (e.g., the different timing of the vegetation period, which is earlier for barley, while wheat is more impacted by the summer

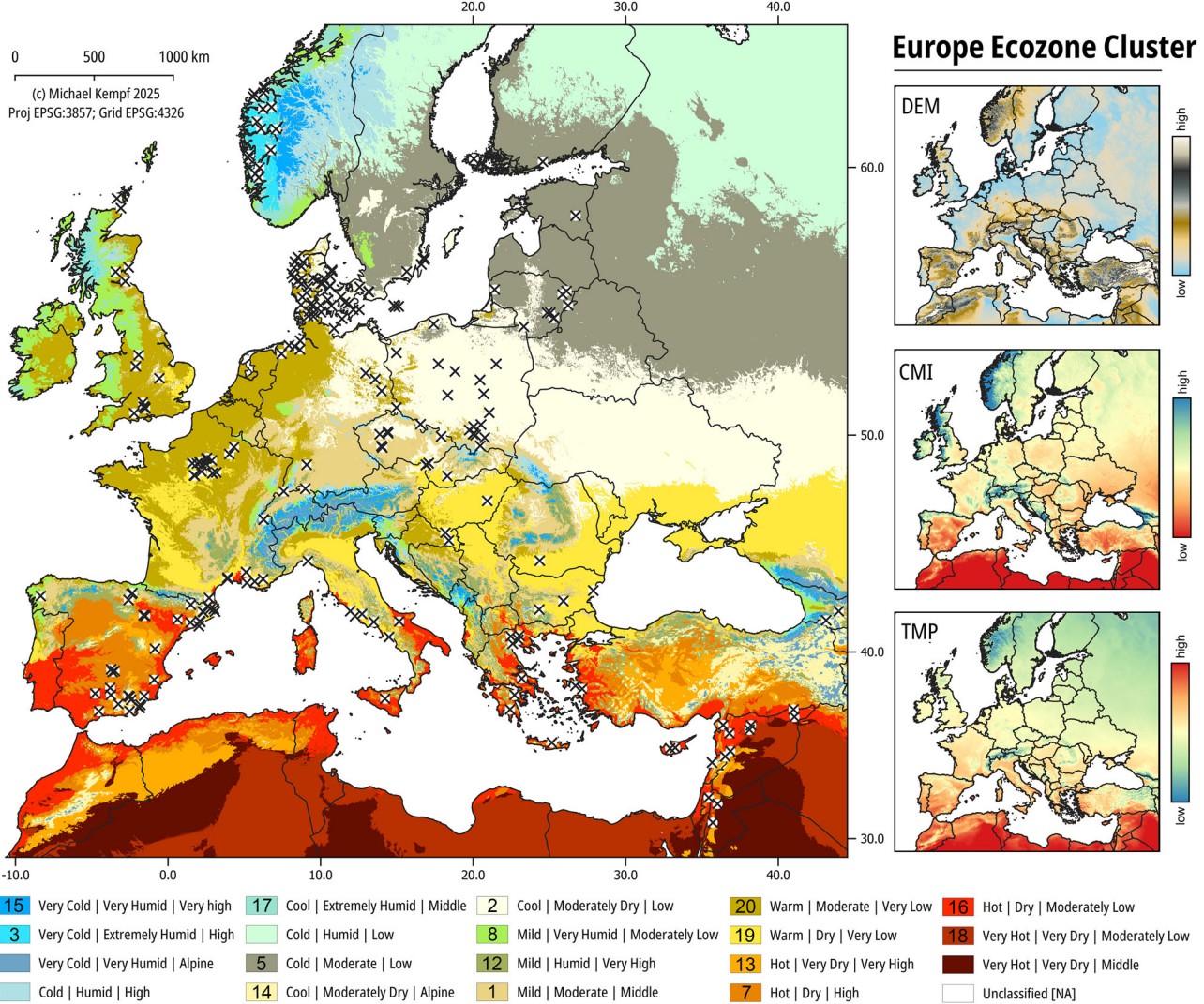

**Fig. 1 | Site distribution over the European Ecozone clusters.** 20 clusters based on temperature (TMP), moisture availability (CWB), and topography (DEM) were defined using *k-means* cluster analysis (with k = 20), including unclassified NA values (i.e., inland water). See the methods and material section for a description of

the open source TMP, CWB and DEM data and their provenance. The sites are distributed over 15 clusters. See Table 1 for the ecozone descriptions and numbering and Fig. S1 for the ecozones displayed without sites. Figure by Michael Kempf, created using the open source R and QGIS software.

**Table 1 | K-means cluster ecozone cluster summary table including description**

| cluster | mean TMP | mean CWB | mean DEM | SD TMP | SD CBW | SD DEM | Description (TMP|CWB|DEM) |
|---|---|---|---|---|---|---|---|
| 1 | 0.429 | 0.405 | 0.205 | 0.039 | 0.017 | 0.021 | Mild|Moderate|Middle |
| 2 | 0.362 | 0.343 | 0.06 | 0.019 | 0.01 | 0.013 | Cool|Moderately Dry|Low |
| 3 | 0.111 | 1 | 0.343 | 0.053 | 0.05 | 0.056 | Very Cold|Extremely Humid|High |
| 4 | 0.116 | 0.429 | 0.065 | 0.02 | 0.007 | 0.015 | Cold|Humid|Low |
| 5 | 0.242 | 0.392 | 0.051 | 0.02 | 0.009 | 0.011 | Cold|Moderate|Low |
| 6 | NA | NA | NA | NA | NA | NA | NA [unclassified|waterbody] |
| 7 | 0.659 | 0.248 | 0.296 | 0.04 | 0.019 | 0.024 | Hot|Dry|High |
| 8 | 0.398 | 0.544 | 0.083 | 0.05 | 0.019 | 0.026 | Mild|Very Humid|Moderately Low |
| 9 | 0.115 | 0.51 | 0.296 | 0.051 | 0.022 | 0.03 | Cold|Humid|High |
| 10 | 0.938 | 0.012 | 0.269 | 0.039 | 0.017 | 0.027 | Very Hot|Very Dry|Middle |
| 11 | 0 | 0.521 | 1 | 0.065 | 0.051 | 0.058 | Very Cold|Very Humid|Alpine |
| 12 | 0.388 | 0.406 | 0.43 | 0.043 | 0.024 | 0.03 | Mild|Humid|Very High |
| 13 | 0.633 | 0.184 | 0.479 | 0.053 | 0.026 | 0.03 | Hot|Very Dry|Very High |
| 14 | 0.329 | 0.318 | 0.718 | 0.051 | 0.025 | 0.043 | Cool|Moderately Dry|Alpine |
| 15 | 0.091 | 0.584 | 0.579 | 0.07 | 0.035 | 0.042 | Very Cold|Very Humid|Very High |
| 16 | 0.799 | 0.226 | 0.092 | 0.032 | 0.021 | 0.027 | Hot|Dry|Moderately Low |
| 17 | 0.297 | 0.749 | 0.131 | 0.052 | 0.031 | 0.038 | Cool|Extremely Humid|Middle |
| 18 | 1 | 0 | 0.078 | 0.031 | 0.018 | 0.023 | Very Hot|Very Dry|Moderately Low |
| 19 | 0.53 | 0.306 | 0.045 | 0.034 | 0.012 | 0.015 | Warm|Dry|Very Low |
| 20 | 0.473 | 0.395 | 0.046 | 0.032 | 0.011 | 0.015 | Warm|Moderate|Very Low |

*TMP* temperature, *CWB* moisture availability, *DEM* digital elevation model (i.e., topography), SD: standard deviation. Cluster 6 represents unclassified water bodies (NA; −99999).

conditions)[28,103] or the environmental diversity of the fields used to grow the different crops[29]. Because human and animal diet is never based on one single crop, the $C_3$ grain sample was kept as one entity for the rest of the analyses. In contrast, the $C_3$ and $C_4$ grains show distinct $\delta^{13}C$ values (Fig. S3C, Supplementary data 2), as expected from their different photosynthetic pathways[13–15]. They are thus considered separately in the rest of the study.

**Temporal isotope variability**
Among the entire $C_3$ grains dataset, there is hardly any evolution of $\delta^{13}C$ over time (linear model [lm] $R^2 = 0.017$, p = 1.75e-17; Pearson's r value = 0.13, p = <2.2e-16; Fig. S3, Supplementary data 2). When distinguishing between the geographical subsets UK (lm $R^2 = 0.044$, p = 0.000276; Pearson's r value = −0.21, p = 0.000276), Southern (lm $R^2 = 0.03$, p = 3.43e-17; Pearson's r value = 0.17, p = <2.2e-16), Central/Western (lm $R^2 = 0.215$, p = 1.80e-32; Pearson's r value = 0.46, p = <2.2e-16) and Northern Europe (including Denmark: lm $R^2 = 0.078$, p = 4.13e-18; Pearson's r value = 0.28, p = <2.2e-16; and excluding Denmark: lm $R^2 = 0.059$, p = 1.55e-09; Pearson's r value = 0.34, p = <2.2e-16), the positive correlation between $C_3$ grain $\delta^{13}C$ values and the grain mean date is particularly weak (Fig. S4A–E, Supplementary data 2). In particular, the slightly stronger relationship observed for Central/Western Europe (Fig. S4B) is biased by the youngest samples from Central France, which exhibit particularly enriched $\delta^{13}C$ values (Fig. S4C). At the ecozone level, a weak to moderate and significant increase in $C_3$ grains $\delta^{13}C$ values can be observed for ecozone 1 (lm $R^2 = 0.359$, p = 5.39e-15; Pearson's r value = 0.60, p = 5.39e-15) and ecozone 17 only (lm $R^2 = 0.223$, p = 0.00157; Pearson's r value = 0.47, p = 1.57e-03) (Figs. 1 and S5, Supplementary data 2). The $C_4$ grains dataset has a small sample size and each geographical area is represented by short chronologies, which does not enable any proper diachronic analysis (Fig. S6A). The slight decrease in $C_4$ $\delta^{13}C$ values over time might thus be only considered statistically significant in Southern Europe despite the chronological gap of nearly a thousand years between the oldest and youngest cluster (Fig. S6B, Supplementary data 2). This implies that the $C_3$ and the $C_4$ grains datasets were

not subdivided into different chronological phases for the subsequent analyses.

**Geographical isotope variability**
Splitting the $C_3$ grain dataset into geographical subsets (UK, Northern, Southern, and Central/Western Europe) shows that the median $\delta^{13}C$ value of $C_3$ grains from Northern Europe is approximately 1‰ lower than that from Southern Europe (Fig. 2A, Table 2). This confirms the previous observations made on different types of samples such as faunal remains[22,23] and modern plants[21]. Yet it has to be stressed that the standard deviation (1 SD) is quite large for both regions (±1.35 and ±1.05, respectively), implying some overlap. Despite the high latitude, $C_3$ grains from the UK exhibit among the highest $\delta^{13}C$ values across time (Figs. 2A and S4), which can be related to the oceanic climate[22,23]. In Denmark, the low $\delta^{13}C$ values of the oldest half of the sample (c. 3700–3000 BCE) shift to particularly enriched $\delta^{13}C$ values for the most recent half of the sample (c. 1000 BCE–1000 CE) (Fig. S7, Supplementary data 2). This might reflect changes in agricultural practices and soil management following its decrease in quality starting from the Neolithic period[30,31].

Using the whole dataset, we observe a significant decrease in $C_3$ grain $\delta^{13}C$ values with increasing latitude (Fig. 2B, Supplementary data 2). From the median values calculated for each latitudinal bin (Table 2, Fig. 2B), the $C_3$ grains $\delta^{13}C$ values from sites above 50° latitude are on average 0.54 to 1.72‰ lower than those of grains from sites at latitudes below 50° (Fig. 2B, Table 2). This confirms the mean offset of around 1–2‰ between Southern and Northern Europe. In comparison, there is a mean variation of 0.46‰ among the median $\delta^{13}C$ values of the latitudinal bins above 50° and approximately 0.33‰ among the median $\delta^{13}C$ values of the bins below 50°. The difference between southern and northern sites is therefore substantial, yet related to an increasing degree of variability towards the north. When excluding the UK and/or Denmark from this dataset due to the overall elevated values in these regions despite their northern latitude, the decrease in $\delta^{13}C$ values with increasing latitude is accordingly even stronger and more significant (Pearson's r value: −0.25 for the whole sample, −0.27 excluding UK and

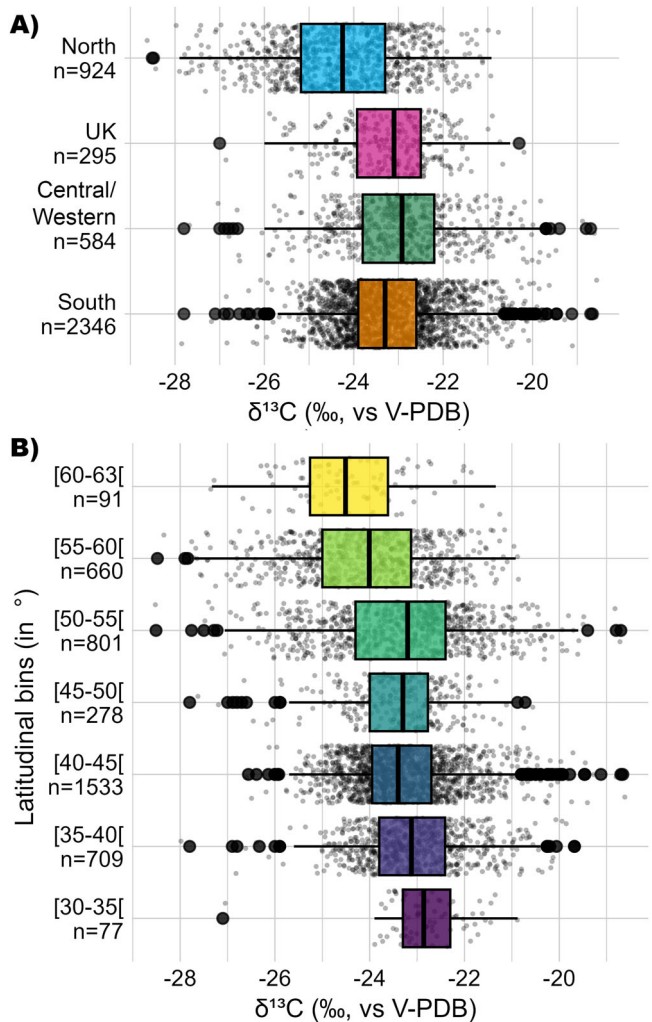

**Fig. 2 | Geographical isotopic variability in C₃ grains. A** C₃ grains δ¹³C values in Europe. **B** C₃ grain δ¹³C variability compared to latitudinal bins within Europe. The middle line of the box represents the median value, the box is delimited by the quartiles Q1 on the left and Q3 on the right and contains the middle half of the sample, the horizontal lines completed by the outlier dots represent the extent of the data. The mean, median, mean absolute deviation (MAD) and standard deviation (1 SD) values for each region and each latitudinal bin are listed in Table 2. The results of the one-way ANOVA tests related to (**A**) and (**B**) and of the Pearson's correlations related to the C₃ grain δ¹³C versus latitude for the whole dataset and for specific subsets are reported in Supplementary data 2. Figure by Margaux L. C. Depaermentier, created using the open source R software.

−0.30 excluding UK and Denmark, with a p-value < 2.2e-16 in each case; see Supplementary data 2).

At the modern country level, the C₃ grains from Lithuania (median −25.18 ± 1.16‰, n = 153) are on average nearly 2.5‰ lower than those from Jordan (median −22.86 ± 0.74‰, n = 46) (Fig. 3A, Table 2) which exceeds the previously defined offset of 1–2‰ between Southern and Northern Europe[21–23]. On the contrary, the Jordan sample is on average 0.80‰ more enriched than those from Greece (median −23.50 ± 0.82‰, n = 383) or Italy (median −23.50 ± 0.97‰, n = 497) despite their shared southern location. Consequently, the North–South-dichotomy is not enough to characterize the different isotopic composition of grains from diverse parts of Europe and does not account for micro-regional environmental diversity. Moreover, the northernmost countries show standard deviations (SD) of 1.29‰ on average (1.12 to 1.67‰ in total), which is sensibly more than in most of the southern (from 0.40 to 1.43‰; mean:

0.92‰) and of the central/western countries (from 0.39 to 1.51‰, mean: 0.90‰) (see the ANOVA test in Supplementary data 2).

Similarly, the C₄ grain δ¹³C values from Lithuania (median: −10.83 ± 0.48‰, n = 20) are on average nearly 1‰ lower than those from Greece (median: −10.19 ± 0.15‰, n = 12), France (median: −10.15‰, n = 16) or Poland (median: −10.19 ± 0.15‰, n = 9) (Fig. 3B, C; Table 2). The sample sizes from Spain (n = 3, median: −10.69 ± 0.15‰) and from the Czech Republic (n = 1) are too low to be considered representative. This trend confirms previous studies from China with depleted C₄ grain δ¹³C values recorded at higher latitudes[25,26]. The pattern is further supported by the larger C₄ grain δ¹³C dataset resulting from Alpha Magnetic Spectrometer (AMS) in Europe[104], showing generally lower δ¹³C values in Northern compared to Southern or Central Europe (Fig. S8). However, AMS stable isotopic data lack precision due to differences in calibration compared to IRMS and provide particularly wide and unusual δ¹³C ranges for C₄ grains[105]. Therefore, these data cannot be used to extend the IRMS dataset in this study. Differences in local plant genotypes are considered more likely triggers for the isotopic variability than climate and water availability[24]. Both C₃ and C₄ grains exhibit lower δ¹³C values in some northern regions relative to the rest of Europe, leading to regional variation in the δ¹³C threshold for identifying C₄ consumption.

## Ecological isotopic variability

The geographical isotopic variability is related to environmental factors captured in the ecozone model. C₃ grain samples from Lithuania (n = 153), Estonia (n = 11), Finland (n = 44), and from parts of Denmark (n = 86) fall into the subhumid temperate lowlands of North-Eastern Europe represented by ecozone 5 (n = 294). Together with ecozone 17 (n = 42)—represented by grains from Norway only—these samples exhibit the lowest median δ¹³C values (−25.01 ± 1.25‰ and −25.08 ± 1.20‰, respectively) for charred C₃ grains in Europe (Fig. 4, Table 2). The high humidity, low temperature and low to moderately elevated topography of these ecozones can explain the depleted δ¹³C values[20]. Ecozone 20 (n = 717), represented by balanced temperate plains scattered over Europe and including samples mostly from Denmark, northern Germany and England, exhibit a much higher median δ¹³C value (−23.00 ± 1.16‰) and its range hardly overlaps with the other northern samples. Beyond the influence of agricultural practices mentioned above[30,31], this can be explained by higher temperatures and moderate humidity characterizing this ecozone. In contrast, the 1007 samples from ecozone 17 and the 140 samples from ecozone 1 show the lowest median δ¹³C values among the sites below 50° latitude (−23.47 ± 0.82‰ and −23.35 ± 1.41‰, respectively). This reflects the mild and moist conditions of these transition zones at a mid-range altitude. In Southern Europe, the ecozones 7 and 13, representing warm highlands in the Mediterranean area, show the highest median δ¹³C values (−22.93 ± 1.35‰ and −22.56 ± 0.87‰, respectively), deriving from the drier and warmer climatic conditions. The ecozones 2, 8 and 19 are scattered over wide areas of Europe and are not related to extreme temperatures. Their isotopic ratios show intermediate values (Table 2). Ecozones 3, 12, 14, 15 and 18 cannot be included in this isotope investigation due to their small sample size.

## Discussion

Building on the substantial geographical and ecological variation in isotope values within C₃ (and to a lesser extent C₄) plants, it is essential to revise the commonly used δ¹³C threshold for identifying C₄ consumption, such as −18.0‰ for mammal collagen (for example, ref. 7). At each investigated site, the C₃ and C₄ grain δ¹³C values from this dataset (Fig. 5A) were used to create theoretical collagen δ¹³C values for a diet based exclusively on these crops (Fig. 5B)—which is no realistic diet for humans or animals and was only used for a first theoretical model. This resulted in site-specific estimations for an overall C₃ grain-based diet with 10% to 20% C₄ grain inputs (Fig. 5B and Supplementary data 1). Our model shows that at several sites from the Baltic and Nordic countries, human or animal collagen δ¹³C values

**Table 2 | Statistical summary for the C$_3$ and C$_4$ grain $\delta^{13}$C values over the latitude bins, regions, modern countries and ecozones**

| Variable | | n C$_3$ grains | Mean C$_3$ grain $\delta^{13}$C (‰) | Median C$_3$ grain $\delta^{13}$C (‰) | MAD for C$_3$ grain $\delta^{13}$C (‰) | 1 SD for C$_3$ grain $\delta^{13}$C (‰) | Comment (C$_3$ grains) | n C$_4$ grains | Mean C$_4$ grain $\delta^{13}$C (‰) | Median C$_4$ grain $\delta^{13}$C (‰) | MAD for C$_4$ grain $\delta^{13}$C (‰) | SD for C$_4$ grain $\delta^{13}$C (‰) | Comment (C$_4$ grains) |
|---|---|---|---|---|---|---|---|---|---|---|---|---|---|
| Latitude bins (in °) | [30–35] | 77 | −22.82 | −22.86 | 0.74 | 0.85 | | 0 | | | | | |
| | [35–40] | 709 | −23.07 | −23.12 | 1.02 | 1.08 | | 0 | | | | | |
| | [40–45] | 1533 | −23.29 | −23.40 | 0.89 | 1.03 | | 18 | −10.45 | −10.49 | 0.44 | 0.31 | |
| | [45–50] | 278 | −23.42 | −23.30 | 0.89 | 1.06 | | 13 | −10.05 | −10.00 | 0.30 | 0.36 | |
| | [50–55] | 801 | −23.31 | −23.20 | 1.44 | 1.47 | | 19 | −10.448421 | −10.29 | 0.42 | 0.73 | |
| | [55–60] | 660 | −24.10 | −24.01 | 1.40 | 1.36 | | 11 | −10.95 | −10.93 | 0.52 | 0.49 | (ends at 55.69°) |
| | [60–63[ | 91 | −24.44 | −24.51 | 1.17 | 1.24 | | 0 | | | | | |
| Region | North | 924 | −24.26 | −24.25 | 1.41 | 1.35 | | 20 | −10.88 | −10.83 | 0.48 | 0.70 | |
| | UK | 295 | −23.21 | −23.10 | 1.04 | 1.06 | | 23 | −10.09 | −10.19 | 0.30 | 0.31 | |
| | Central/ West | 584 | −22.95 | −22.92 | 1.16 | 1.30 | | 0 | | | | | |
| | South | 2346 | −23.21 | −23.30 | 0.95 | 1.05 | | 18 | −10.45 | −10.49 | 0.44 | 0.31 | |
| Modern country | Andorra | 3 | −22.33 | −22.10 | 0.37 | 0.62 | small sample size | 0 | | | | | |
| | Bulgaria | 22 | −23.45 | −23.50 | 0.44 | 0.40 | | 0 | | | | | |
| | Croatia | 27 | −23.64 | −23.80 | 0.59 | 0.88 | | 0 | | | | | |
| | Cyprus | 8 | −22.97 | −23.09 | 0.76 | 0.85 | small sample size | 0 | | | | | |
| | Czech Republic | 67 | −23.14 | −23.00 | 0.74 | 0.73 | | 1 | −9.90 | −9.90 | 0.00 | NA | small sample size |
| | Denmark | 323 | −23.56 | −23.36 | 1.19 | 1.32 | | 0 | | | | | |
| | England | 259 | −23.14 | −23.02 | 0.93 | 1.01 | | 0 | | | | | |
| | Estonia | 11 | −24.47 | −24.50 | 1.93 | 1.67 | rather small sample size | 0 | | | | | |
| | Finland | 44 | −24.14 | −24.39 | 1.05 | 1.23 | | 16 | −10.19 | −10.15 | 0.36 | 0.44 | |
| | France | 206 | −23.17 | −23.20 | 0.82 | 0.92 | | 0 | | | | | |
| | Georgia | 20 | −21.87 | −21.60 | 1.11 | 1.43 | | 0 | | | | | |
| | Germany | 302 | −22.62 | −22.60 | 1.33 | 1.51 | | 0 | | | | | |
| | Greece | 383 | −23.50 | −23.50 | 0.76 | 0.82 | | 12 | −10.32 | −10.19 | 0.15 | 0.27 | |
| | Hungary | 4 | −24.20 | −24.25 | 0.15 | 0.22 | small sample size | 0 | | | | | |
| | Italy | 497 | −23.38 | −23.50 | 0.89 | 0.97 | | 0 | | | | | |
| | Jordan | 46 | −22.66 | −22.86 | 0.82 | 0.74 | | 0 | | | | | |
| | Lebanon | 2 | −25.45 | −25.45 | 2.45 | 2.33 | small sample size | 0 | | | | | |
| | Lithuania | 153 | −25.13 | −25.18 | 1.01 | 1.16 | | 20 | −10.88 | −10.83 | 0.48 | 0.70 | |
| | Norway | 63 | −24.82 | −24.85 | 1.33 | 1.22 | | 0 | | | | | |
| | Poland | 52 | −23.84 | −23.90 | 1.12 | 0.94 | | 9 | −10.16 | −10.19 | 0.15 | 0.25 | small sample size |
| | Scotland | 36 | −23.73 | −23.90 | 1.48 | 1.24 | | 0 | | | | | |
| | Slovakia | 10 | −23.66 | −23.50 | 0.44 | 0.39 | rather small sample size | 0 | | | | | |
| | Spain | 965 | −23.08 | −23.22 | 1.08 | 1.17 | | 3 | −10.66 | −10.69 | 0.15 | 0.15 | small sample size |
| | Sweden | 327 | −24.46 | −24.50 | 1.19 | 1.12 | | 0 | | | | | |
| | Switzerland | 7 | −23.97 | −24.10 | 0.30 | 0.46 | small sample size | 0 | | | | | |
| | Syria | 144 | −23.07 | −22.98 | 0.77 | 1.02 | | 0 | | | | | |
| | Turkey | 168 | −23.11 | −23.12 | 0.79 | 0.83 | | 0 | | | | | |
| Ecozone cluster | 1 | 140 | −23.48 | −23.35 | 1.42 | 1.41 | | 8 | −10.34 | −10.44 | 0.37 | 0.49 | small sample size |
| | 2 | 511 | −23.79 | −23.80 | 1.33 | 1.41 | | 10 | −10.13 | −10.19 | 0.15 | 0.25 | rather small sample size |

**Table 2 (continued) | Statistical summary for the $C_3$ and $C_4$ grain $\delta^{13}C$ values over the latitude bins, regions, modern countries and ecozones**

| Variable | | n $C_3$ grains | Mean $C_3$ grain $\delta^{13}C$ (‰) | Median $C_3$ grain $\delta^{13}C$ (‰) | MAD for $C_3$ grain $\delta^{13}C$ (‰) | 1 SD for $C_3$ grain $\delta^{13}C$ (‰) | Comment ($C_3$ grains) | n $C_4$ grains | Mean $C_4$ grain $\delta^{13}C$ (‰) | Median $C_4$ grain $\delta^{13}C$ (‰) | MAD for $C_4$ grain $\delta^{13}C$ (‰) | SD for $C_4$ grain $\delta^{13}C$ (‰) | Comment ($C_4$ grains) |
|---|---|---|---|---|---|---|---|---|---|---|---|---|---|
| | 3 | 4 | −24.62 | −24.69 | 1.25 | 1.07 | small sample size | 0 | | | | | |
| | 5 | 294 | −24.96 | −25.01 | 1.09 | 1.25 | | 20 | −10.88 | −10.83 | 0.48 | 0.70 | only Lithuania |
| | 7 | 338 | −22.83 | −22.93 | 1.22 | 1.35 | | 0 | | | | | |
| | 8 | 29 | −24.18 | −23.76 | 0.83 | 1.17 | | 0 | | | | | |
| | 12 | 3 | −22.33 | −22.10 | 0.37 | 0.62 | small sample size | 0 | | | | | |
| | 13 | 86 | −22.62 | −22.56 | 0.94 | 0.87 | | 0 | | | | | |
| | 14 | 6 | −21.32 | −21.30 | 0.67 | 0.64 | small sample size | 0 | | | | | |
| | 15 | 5 | −20.88 | −21.10 | 1.04 | 1.50 | small sample size | 0 | | | | | |
| | 16 | 964 | −23.15 | −23.20 | 0.95 | 1.01 | | 12 | −10.32 | −10.19 | 0.15 | 0.27 | rather small sample size |
| | 17 | 42 | −24.97 | −25.08 | 1.48 | 1.20 | | 0 | | | | | |
| | 18 | 3 | −21.97 | −22.10 | 0.15 | 0.32 | small sample size | 0 | | | | | |
| | 19 | 1007 | −23.47 | −23.50 | 0.74 | 0.82 | | 11 | −10.21 | −10.10 | 0.30 | 0.42 | rather small sample size |
| | 20 | 717 | −23.03 | −23.00 | 1.05 | 1.16 | | 0 | | | | | |

The values indicated for small sample sizes are considered non-representative for the related category.

below −19.0‰ (with a mean SD of 0.59‰)—and up to −19.68 ± 0.94‰ at Bėlis lake, Lithuania (n = 10), for example—already reflect a low $C_4$ input within a primarily $C_3$-grain-based diet. In the Mediterranean, the same $C_4$ input would result in collagen $\delta^{13}C$ values above −17.0 ± 0.77 or −16.0 ± 0.57‰ (Fig. 5B and Supplementary data 1).

It is generally accepted that at least 20% of dietary protein from an alternative source (such as $C_4$ compared to $C_3$ crops) is required to be detected in collagen[106]. However, with a model based on theoretical grain-based diets, a 20% $C_4$ input produces excessively elevated $\delta^{13}C$ estimates for mammal collagen (Supplementary data 1). This highlights the limits of a model based on non-realistic grain-based diets, as despite the important role of grains in human diet, both humans and animals have more varied diets in reality—the latter even consuming mostly other parts of the plant and only rarely grains. And with collagen $\delta^{13}C$ values reflecting the protein component of the diet[16], plant intake contributes less to collagen isotopic composition than animal-derived proteins, which may alter this threshold value in omnivorous diets. Moreover, in regions where the diet includes significant proportions of mushrooms[107], forest-derived foods influenced by the canopy effect[108,109], and/or freshwater fish[110–112], consumers are exposed to more depleted $\delta^{13}C$ values. These foods have a much stronger influence on collagen $\delta^{13}C$ values than any enrichment from $C_4$ plants. And since such conditions are highly plausible in the Baltics and in Scandinavia, the threshold $\delta^{13}C$ value for $C_4$ consumption might be even lower than those suggested by the model with 10% $C_4$ input (Figs. 5A, B and 6). On the contrary, potential marine food consumption[17] or sea-spray effects in coastal areas[113] need to be considered to avoid any over-estimation of $C_4$ plant intake.

A model involving all other food resources would go beyond the scope of this paper. But to accurately estimate the actual local threshold values for $C_4$ consumption, it is essential to use $\delta^{13}C$ and $\delta^{15}N$ values from as many local and contemporaneous food resources as possible, including crops (for example[32]). Wild plants would further represent a better baseline for herbivore's diets. Yet these are much more seldom in archaeological remains and their isotopic ratios are hardly represented in bioarchaeological studies so far. A model based on modern plants[21] or on tree $\delta^{13}C$ values[114] can thus further serve as comparison $\delta^{13}C$ baseline for herbivore's diets. The combination of various isotope systems[113,115,116] and/or the application of mixing models[117] are further powerful approaches to disentangle the diverse dietary sources.

Considerable isotopic variability is observed within each site, region and ecozone (Figs. 2–4 and 5A, Supplementary data 1–2), indicating that a range rather than a sharp threshold value is more appropriate (Figs. 5B and 6B). This isotopic variability is not only related to the gradual and complex variation of environmental settings across Europe, but is also intrinsic to the grains, as grain $\delta^{13}C$ may vary for up to 0.5‰ within one ear despite same species and same growing conditions[33]. Differing growing conditions, in particular various watering practices, can further impact local grain $\delta^{13}C$ values by up to 1.7‰[24,28]. Wild plants $\delta^{13}C$ values may be good comparison references to disentangle anthropogenic and natural differences in water regimes[21,118,119]. Another variability might be induced by diverging analytical uncertainty between datasets, as the data result from different laboratories and protocols and were obtained with different calibrations—which are relevant information to make sure that the datasets are comparable[120]. Unfortunately, information on calibration, precision and accuracy of the published isotope data were mostly missing, thus it was not possible to assess the impact on the presented results. Yet when available, the error value for replicated samples was very low and still within the range of analytical errors.

An evolution of the grain $\delta^{13}C$ values might be expected over time as well, especially when considering the various climatic phases that implied a great variation in moisture and temperature throughout Europe over the investigated time frame. The presented regional differences in grain $\delta^{13}C$ values have therefore varied between the climatic phases (Fig. S10), yet the isotopic variations within each region over the various climatic phases is overall significantly weak (Supplementary data 2). A larger dataset for the most recent periods might reveal more effective trends. In this study, the

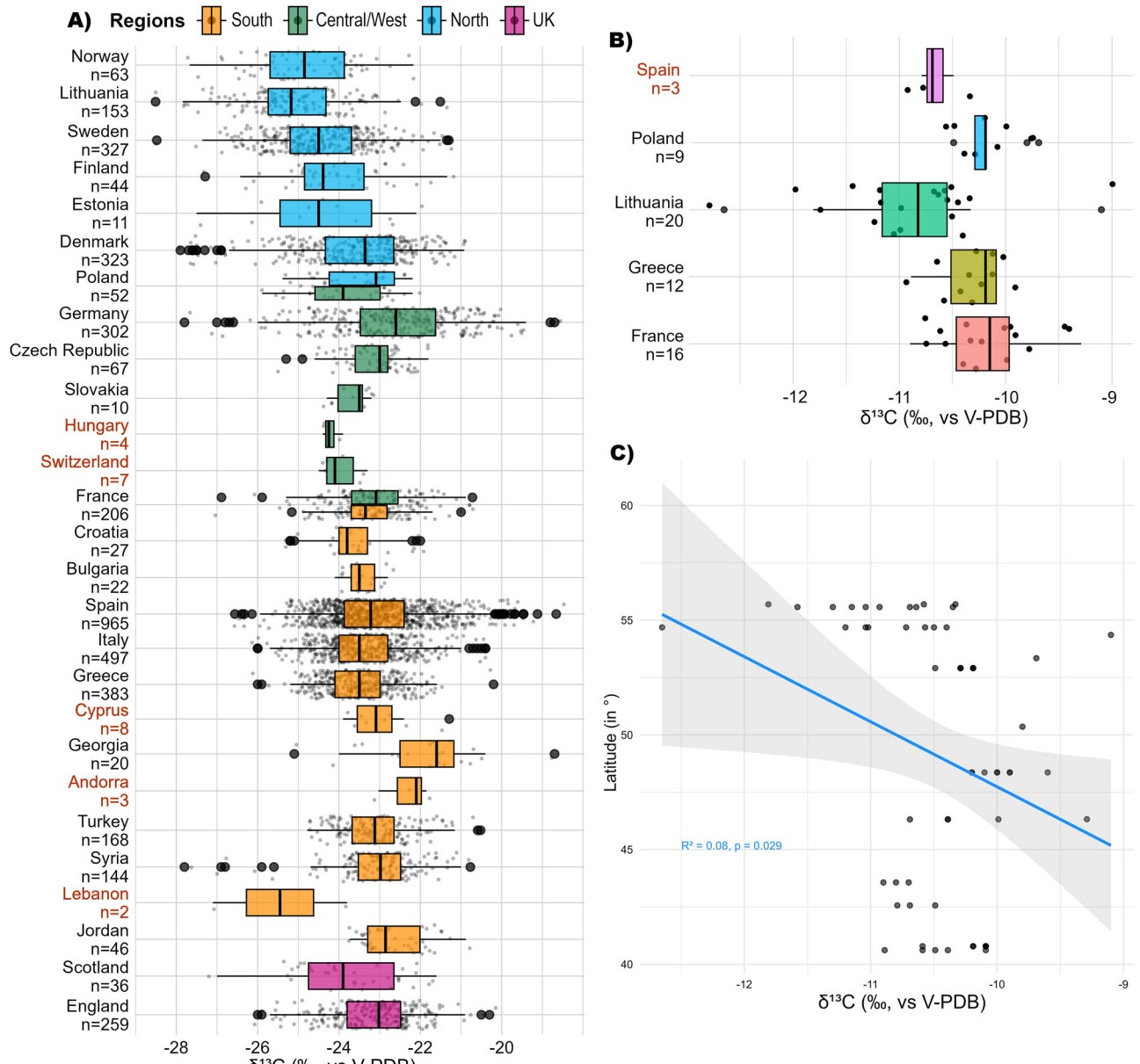

**Fig. 3 | Differences in archaeological charred $C_3$ and $C_4$ grain $\delta^{13}C$ values in Europe. A** $C_3$ grain $\delta^{13}C$ values over the modern countries. **B** $C_4$ grain $\delta^{13}C$ values over modern countries. **C** The comparison between latitude and $C_4$ grain $\delta^{13}C$ values shows a weak but significant negative correlation. Boxplots are defined in Fig. 2. The mean, median, MAD and SD values for each modern country are listed in Table 2.

The results of the related one-way ANOVA tests and Pearson's correlations are available from Supplementary data 2. The red labels show non-representative sample sizes (n < 10). Figure by Margaux L. C. Depaermentier, created using the open source R software.

---

impact of the chronological depth and imbalance covered by this dataset can be considered weak (Figs. S2–S5 and S10).

The isotopic difference reported in this study between regions and ecozone therefore remains strong enough to highlight environmentally-driven differences (Figs. 2–7, Table 2) even from grains that were possibly undergoing various cultivation practices. Our approach thus demonstrates that the threshold at −18.0‰ for consumer's collagen is not universal and is mostly valid in the ecozone 2 (−17.51 ± 1.28‰), whereas the northern ecozone 5 and the widespread ecozone 17 require a threshold $\delta^{13}C$ value closer to −19.0‰, i.e., −18.59 ± 1.12 and −18.67 ± 1.08‰, respectively. In high-temperature Mediterranean lowlands (ecozone 16), European plains (ecozone 19), and Atlantic regions (ecozone 20), the threshold shifts to −17.0‰ (i.e., −16.88 ± 0.90‰, −17.15 ± 0.74‰, and −16.71 ± 1.05‰, respectively; Figs. 6, 7, Tab. S1). In the arid southern regions (ecozone 13 [−16.31 ± 0.78‰], and partially ecozones 7 and 16), it approaches −16.0‰

(Fig. 7, Tab. S1). Yet in each case, the SD ranges from 0.74 to 1.26‰, stressing again the isotopic variability within ecozones.

To conclude, this paper offers both a European-wide $\delta^{13}C$ baseline from archaeological charred $C_3$ grains and a threshold-value-model for environmentally adjusted identification of $C_4$ consumption from the site to the ecozone level across Europe (Fig. 7, Tab. S1). The grain baseline offers the advantage to consider a fundamental dietary resource (in particular for humans) as reference data and can be completed by local foods resources at the site level for more holistic and accurate interpretations. A comparison to wild proxies further enhances the results for animal diets. The threshold estimations are particularly suitable in bioarchaeological, ecological or palaeontological studies for which local plants or food resources are unavailable for calculating an isotopic baseline. However, it requires to account for the great degree of isotopic variability within each geographical entity as underlined by the SD values—a variability slightly increasing with latitude.

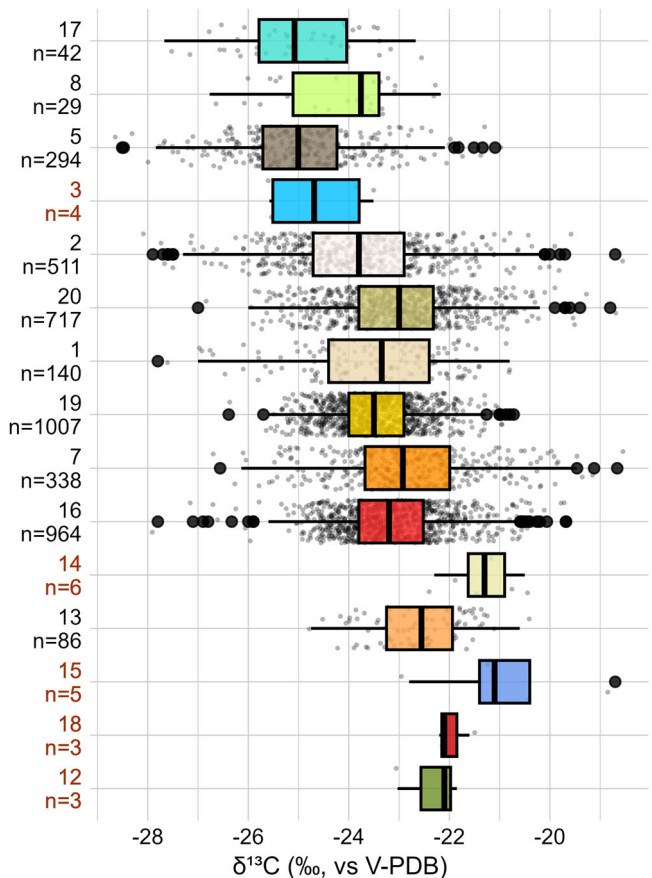

**Fig. 4 | C$_3$ grains $\delta^{13}$C values over the European ecozones clusters.** The ecozone numbers refer to the numbering in Table 1. Boxplots are defined in Fig. 2. The boxplots are ordered from top to bottom according to decreasing latitude and to increasing temperature within the ecozone. The red labels underline non-representative sample sizes (n < 10). The mean, median, MAD and SD values for each ecozone can be found in Table 2. The results of the one-way ANOVA test are available from Supplementary data 2. Figure by Margaux L. C. Depaermentier, created using the open source R software.

In this context, the point-based approach (Fig. 5) provides more accurate yet geographically discrete data, while the interpolation-based approach (Fig. 6) offers ecologically sensitive estimates over large areas at the ecozone level, both related to some degree of uncertainty or variability. Future datasets could be used to test (and if necessary adjust) the interpolated values in areas of currently low site density. This innovative and context-sensitive ecozone clustering model based on temperature, humidity, and elevation thus enables more accurate interpretations of both animal ecologies and anthropogenic social and agricultural dynamics across Europe by avoiding over- or underestimation of C$_4$ consumption.

## Methods and material
### Isotopic dataset
The material used in this study consists of published $\delta^{13}$C values from charred grains derived from archaeological context, compiled into one single dataset[27]. The data was collected from 75 publications until September 2025[28–101], also using the open access online repositories IsoArch[121], MAIA[122], Isotòpia[123], IsoMedIta[124] and CIMA[125]. In total, this represents 4,210 $\delta^{13}$C values of C$_3$ and C$_4$ grains derived from 260 sites dated between 8000 BCE and 1800 CE. The represented C$_3$ plants are oat (*Avena species*, n = 58), rye (*Secale species*, n = 325), barley (*Hordeum species*, n = 1843), and wheat (*Triticum species*, n = 1923). Broomcorn millet (*Panicum miliaceum*, n = 57) and foxtail millet (*Setaria italica*, n = 4) represent the C$_4$ crops from this dataset. To facilitate visualization and pattern-recognition, the C$_3$ and

C$_4$ grain datasets were considered separately for the various analyses due to their distinct $\delta^{13}$C values and due to the particularly small C$_4$ grain sample size. C$_3$ and C$_4$ grain $\delta^{13}$C values were then combined to address the question of identifying the introduction of C$_4$ crops in C$_3$ plants-based diets. Despite the fact that anthropogenic agricultural practices such as irrigation can impact grain $\delta^{13}$C values[24,28], altering the natural and ecological signal, crops remain an important proxy for human diet regardless of agricultural practices, as their isotopic composition would be transferred to human tissues all the same. The ecological differences are considered important enough across Europe to be detectable from grain isotopic composition despite anthropogenic alterations. Wild plants would have represented a better proxy for animal diet, however, this proxy is lacking from archaeological contexts—or could be derived from tree $\delta^{13}$C values for the most recent periods[114].

Geographically, the research area spans modern Europe and the Mediterranean countries of the Near East, i.e., between 30 and 63° (N) latitude and between −8 and 45° (E) longitude. Yet the data is not evenly distributed and Denmark is over-represented in terms of sites, while Greece is over-represented in terms of number of samples. There are considerable gaps in several regions of Europe (see Fig. 1 and the isotopic dataset[27]). Chronologically, this dataset covers most archaeological and historical periods and ranges from 8000 BCE to 1800 CE. In this context, it is important to stress that the oldest samples (8000–6000 BCE) exclusively originate from Greece, whereas samples from Northern Europe are predominantly younger than 1000 BCE—except for some larger site samples dated between 4000 BCE and 2000 BCE. Modern data created in the framework of experimental archaeology were not included in the dataset because of the controlled conditions in which they were produced and because of the different present-day atmospheric composition compared to pre-industrial periods[126].

Only data obtained from Isotope-Ratio Mass Spectrometry (IRMS) were selected for analyses. Despite the fact that these values are not comparable to IRMS isotopic values due to different calibrations[105], a European-wide dataset obtained from Accelerator Mass Spectrometry (AMS) in the context of radiocarbon measurements[104] was also used as a comparison dataset to verify whether the same trend is visible among both datasets. Importantly, grain $\delta^{13}$C values are sometimes (i.e., in 12 out of 64 publications used in this study) published in form of corrected values to consider the charring effect on the carbon isotope composition of archaeological C$_3$[127,128] and C$_4$ grains[129,130]. But following the approach by Gron et al.[30], we are considering here only uncorrected values in order to enhance the comparability between datasets. This is considered to have no significant impact on this study's results, as the charring effect is not systematic[33] particularly low on grain $\delta^{13}$C values (i.e., 0.06 to 0.18‰[33,127–130] on C$_3$ and C$_4$ grains for a heat up to 300 °C) and remains below analytical errors for isotope analyses[131].

### Statistical analyses
All statistical analyses were performed using R software[132] and the results and relevant values are summarised in Supplementary data 2. To determine the relationship between grain $\delta^{13}$C values and chronology or geographical location, we applied both Pearson's correlation tests and linear models—the latter using the *lme4* package[133]. The Pearson correlation coefficient (Pearson's r value) indicated the strength and direction of a linear relationship between two tested variables, the confidence interval gives the uncertainty range for the true correlation. While in the linear model, the R-squared values show the percentage of the dataset affected by the relationship and the p-values show the significance of the results. To determine the geographical scale at which significant changes in grain values occur, the dataset was divided into various bins. At the largest scale, the main regions of Europe, split into Northern Europe, Southern Europe, Central/Western Europe and the UK. This is not only convenient but also follows expectations based on previous research[21–23]. At the smallest geographical scale, the dataset was binned according to the borders of modern countries. Boxplots were used for data visualization and one-way ANOVA tests were performed to test the

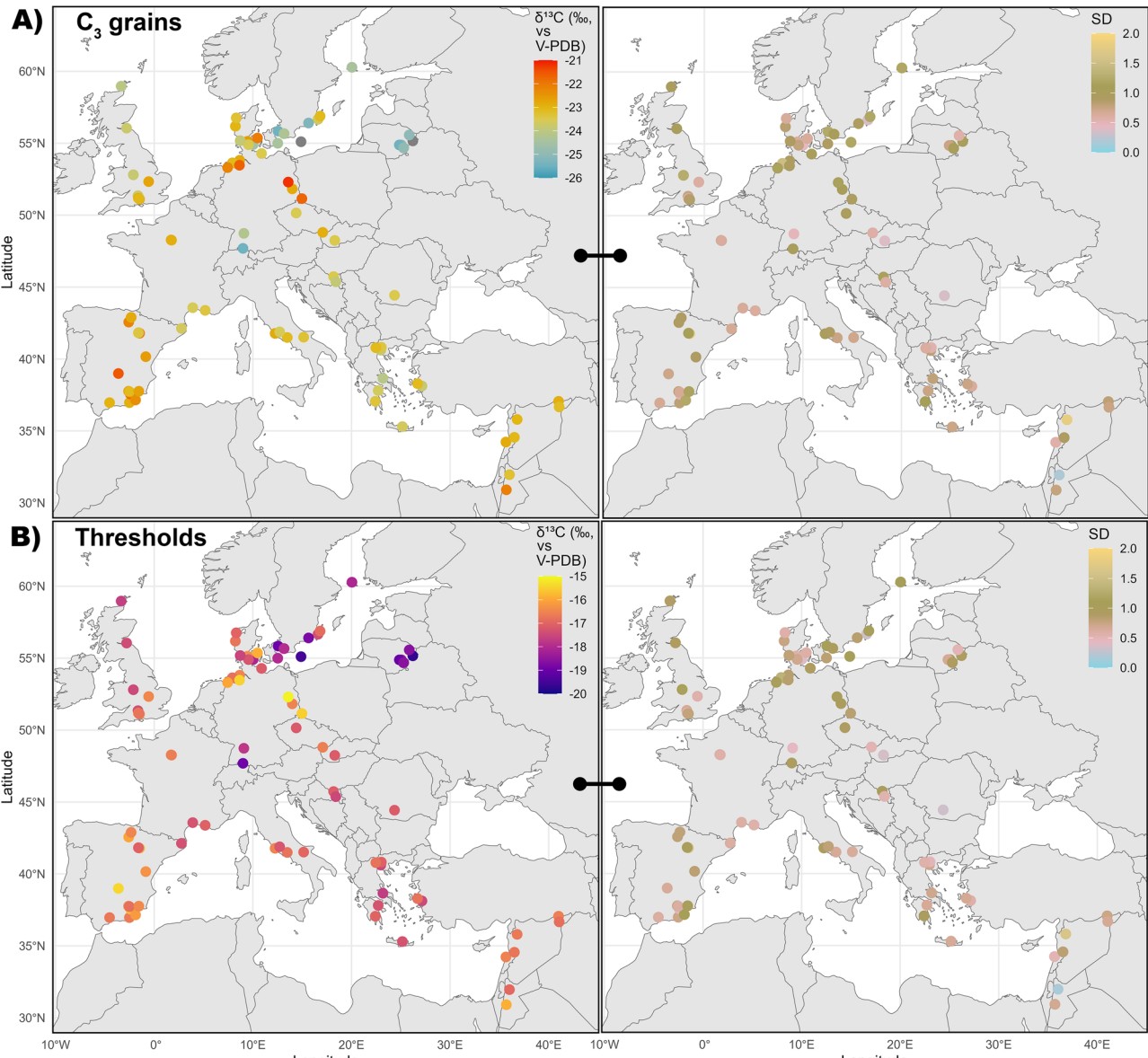

**Fig. 5 | Point-based approach for baseline C₃ grain δ¹³C values and estimated threshold values for C₄ diet identification in mammal collagen at sites with n ≥ 10 grains. A** Median C₃ grain δ¹³C values (left) and related SD (right). **B** Median estimated threshold δ¹³C values for mammal collagen (left) and related SD (right) based on a theoretical 100%-grain-based-diet. The mean, median, SD and MAD values for each site are listed in Supplementary data 1. The same maps including even site with n < 10 are in Fig. S9. Figure by Margaux L. C. Depaermentier, created using the open source R software.

difference in grain δ¹³C values between the investigated clusters. The results and relevant values for the ANOVA tests are summarised in Supplementary data 2). Because our dataset shows a particularly important isotopic variability, the results for each considered bin or entity/group are presented in the text using the median value (since the mean value is more sensitive to extreme outliers) and the related one standard deviation (1 SD). The tables are showing the mean, median, median absolute deviation (MAD) and 1 SD for each category/bin. In order to integrate an ecological dimension to the investigation of grain δ¹³C variability, the dataset was also binned into newly determined ecozones, as presented in the section below.

### Environmental cluster
For the cluster analysis, we used these R-packages: *terra*[134], *sf*[135,136], *gtools*[137], *dplyr*[138] and *ggplot2*[139], *ggspatial*[140], *gridExtra*[141] for plotting, as described in the R-code provided in the open access repository to this paper[142]. We used an unsupervised *k*-means clustering approach based on three spatial predictors to differentiate environmental ecozones across Europe. Components

include elevation (DEM), mean temperature (T), and the Climatic Moisture Index (CMI). A DEM derived from the USGS (United States Geological Survey, Global Multi-resolution Terrain Elevation Data 2010, https://earthexplorer.usgs.gov/; last accessed 19th of June 2025). Monthly resolved climate variables for the period 1980-2018 were downloaded from CHELSA[143]. The CMI represents a standardized water availability index, calculated as the ratio of precipitation (P) to potential evapotranspiration (PET) with $CMI = P - PET$. We used CHELSA v2.1 monthly CMI data based on the Penman-Monteith equation for PET and downscaled from ERA5 reanalysis. The grids were reprojected to a meter-based projection (EPSG:3857) and cropped to the extent of the geographic European landmass. The dataset (n = 468) were aggregated to a 1000 m resolution using bilinear interpolation prior to clustering. Monthly layers were averaged to create a multiannual mean (1980–2018). To allow comparison between variables with different scales and units, all raster layers were standardized using z-score normalization with mean and standard deviation (sd) of the raster ($z = $ cell value $-$ mean_raster / sd_raster). A regular grid of 1 km

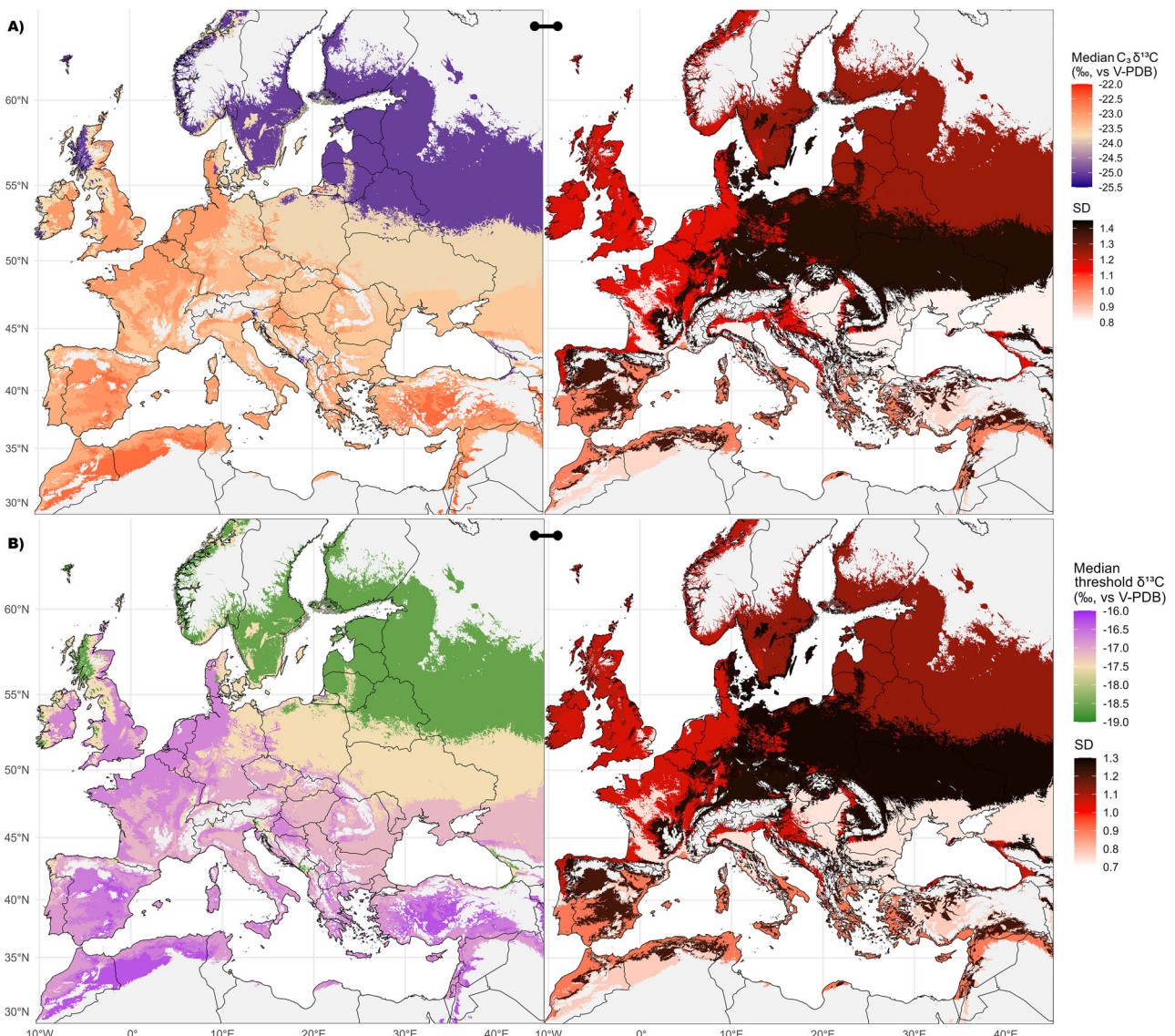

**Fig. 6 | Ecozone-based interpolation of the $C_3$ grain $\delta^{13}C$ baseline and of the estimated threshold $\delta^{13}C$ values for detecting $C_4$ consumers. A** Median $\delta^{13}C$ (left) and SD values (right) of charred $C_3$ grains interpolated at the ecozone level. **B** Median $\delta^{13}C$ (left) and SD values (right) of the estimated threshold ranges for $C_4$ consumption for each ecozone. Ecozones 3, 12, 14, 15 and 18 are left grey due to their too small sample sizes. Ecozones 4, 9, 10, 11 and 15 are left grey due to the absence of data. The ecozone's mean, median, SD and MAD values for $C_3$ grains and for $C_4$ consumption are listed in Table 2 and Tab. S1, respectively. Figure by Michael Kempf, created using the open source R and QGIS software.

spacing was generated across the study area and the centroid of each grid cell was calculated for regular point sampling. To reduce edge effects and avoid NA values near the coast, centroids were restricted to land areas using a simplified buffer around the European landmass boundaries. At each centroid location, the values were extracted from the normalized rasters.

### *K-Means* clustering

*K-means* clustering was performed using the extracted values, with the determined number of clusters $k = 20$. This dimensionality was chosen to balance regional ecological resolution with model interpretability, including NA values (replaced by $-99999$ during the cluster analysis to protect correct geographical raster reassignment). Each centroid was assigned a cluster label based on the combined environmental profile of elevation, temperature, and moisture availability. Cluster labels were spatially joined to centroid coordinates and rasterized back into a continuous spatial layer using the DEM grid as a template. The resulting map classifies observational sites into discrete clustered ecozones (Fig. 1).

To characterize each of the 20 ecozone clusters based on the data variability, we summarized the environmental properties of each cluster using the mean and standard deviation of the three input variables: Elevation (ELEV), T, and CMI. All values were normalized using z-scores prior to clustering, enabling direct comparison across variables. The clusters were then qualitatively interpreted based on their relative environmental signatures. For example, clusters with low temperatures, moderately dry conditions and high elevations were categorized as Cool|Moderately Dry| Alpine zones. Clusters with high temperatures and low moisture availability in low areas were described as Very Hot|Very Dry|Moderately Low. Intermediate clusters were labeled based on transitional or temperate climate conditions (Table 1).

### Determining consumer's $\delta^{13}C$ values

To determine the theoretically expected $\delta^{13}C$ values of consumer tissues from a mixed $C_3$-$C_4$ diet, we first associated each $C_3$ grain $\delta^{13}C$ value to a measured or assumed $C_4$ grain $\delta^{13}C$ value. This means that for each site at

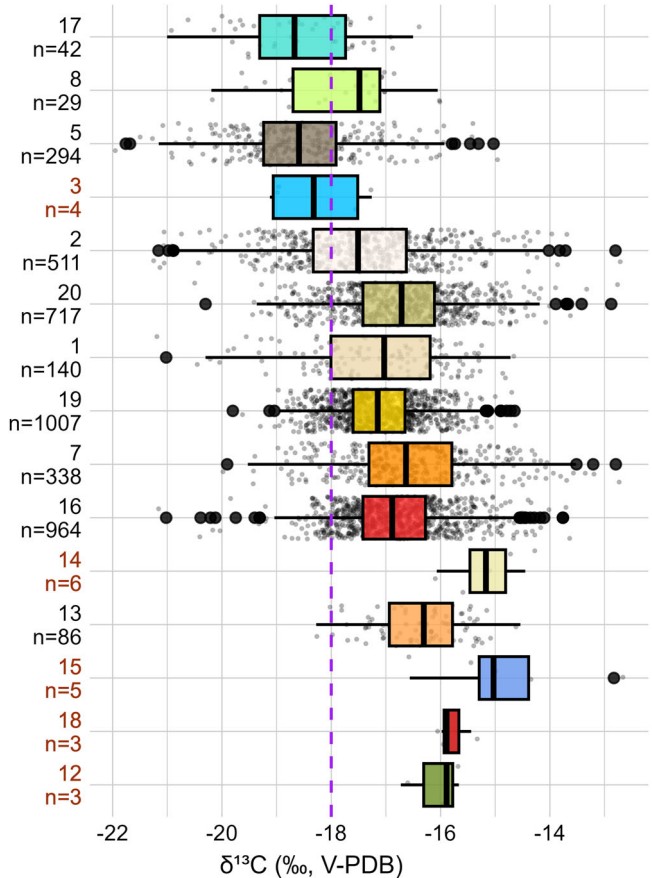

**Fig. 7 | Theoretical $\delta^{13}$C threshold ranges for C$_4$ consumption across the European ecozone clusters.** The ecozone numbers refer to the numbering in Table 1. Boxplots are defined in Fig. 1. The boxplots are ordered from top to bottom according to decreasing latitude and to increasing temperature within the ecozone. The red labels underline non-representative sample sizes (n < 10). The mean, median, MAD and SD values for each ecozone are listed in Tab. S1. The purple line represents the revised $\delta^{13}$C threshold value for C$_4$ consumption in mammal collagen (i.e., −18.0‰). Figure by Margaux L. C. Depaermentier, created using the open source R software.

which $\delta^{13}$C values of both C$_3$ and C$_4$ grains were available, each C$_3$ grain $\delta^{13}$C value got associated with the mean C$_4$ grain $\delta^{13}$C value of the site. However, most of the sites included in this study provided no C$_4$ grain. In this case, a theoretical C$_4$ grain $\delta^{13}$C value was determined for the site based on the observed values from this dataset. In most regions of Europe, the C$_4$ grain $\delta^{13}$C value is thus expected to be −10‰[32,72,87]. Yet we observed that the C$_4$ grain $\delta^{13}$C values in Lithuania—and by extension presumably in the northernmost latitudes of Europe—were rather around −11‰[68]. Similarly, C$_4$ grain $\delta^{13}$C values from the western Mediterranean area seem to be closer to −10.5‰[35,47]. These regional values were thus used as theoretical C$_4$ grain $\delta^{13}$C values associated with the measured C$_3$ grain $\delta^{13}$C values at each site lacking C$_4$ grains (see summary in Supplementary data 1).

In a second step, 5‰ was added to each measured or theoretical grain $\delta^{13}$C value to mimic the fractionation offset that applies between the diet and the consumer's collagenous tissues after consumption[16,17]. We applied this to both C$_3$ and C$_4$ grains, resulting in theoretical end-members collagen $\delta^{13}$C values for 100% C$_3$ and 100% C$_4$ based diet, respectively. In a third step, we used these end-members values for each grain to create theoretical collagen $\delta^{13}$C values for a C$_3$-grain-based diet including either 10% or 20% C$_4$ grains. These three first steps were done at the grain level to minimize the loss of resolution and information when working with mean values. In a fourth step, we eventually calculated a mean $\delta^{13}$C value for these two types of diet at

the site level (Supplementary data 1). It is fundamental to note that these fictitious diets based on 100% grains are not existing in nature and only represent a theoretical model using grains only.

Because a 100% grain-based diet does not exist in nature, the model using 10% of C$_4$ input was considered the most reliable basis for estimating the related collagen $\delta^{13}$C values with low C$_4$ input in a normal mixed diet. These results are presented at the site-level to account for local variability in threshold $\delta^{13}$C values for C$_4$ consumption (Fig. 5). The ecozone cluster model was used to create a map of interpolated threshold $\delta^{13}$C values for C$_4$ consumption and hence suggest environmentally adjusted threshold $\delta^{13}$C values for C$_4$ consumption (Fig. 6C). The European background maps used to create Figs. 5 and 6 are vector map data from https://www.naturalearthdata.com/, implemented using the package *rnaturalearth* in R-Software[132,144].

## Materials & correspondence

The corresponding authors are MLCD and MK. The isotopic dataset used in this study is available from the open access repository: Depaermentier, M. L. C. (2025). Isotopic Dataset to: Depaermentier, MLC, Kempf, M, Motuzaitė Matuzevičiūtė, G. "Environmentally adjusted $\delta^{13}$C thresholds for accurate detection of C$_4$ plant consumption in Europe" [Data set]. In Communications Earth & Environment. Zenodo. https://doi.org/10.5281/zenodo.17571650 [ref. 27 in this paper]. The data to reproduce the ecozone clusters is available from this open access repository: Kempf, M. (2025): Related files to: Depaermentier, MLC; Kempf, M; Motuzaitė Matuzevičiūtė, G: Environmentally adjusted $\delta^{13}$C thresholds for accurate detection of C$_4$ plant consumption in Europe (2025) [Data set]. Zenodo. https://doi.org/10.5281/zenodo.15695070 [ref. 147 in this paper]. Climate variables used in this article are freely available from Karger et al. (2017): https://chelsa-climate.org/ (last accessed 19th of June 2025) [ref. 143 in this paper]. The Digital Elevation Model (DEM) can be downloaded from the USGS earthexplorer server: https://earthexplorer.usgs.gov/, last accessed 19th of June 2025.

## Reporting summary

Further information on research design is available in the Nature Portfolio Reporting Summary linked to this article.

## Data availability

The compiled isotopic dataset used in this study is available from the open access repository: Depaermentier, M. L. C. (2025). Isotopic Dataset to: Depaermentier, MLC, Kempf, M, Motuzaitė Matuzevičiūtė, G. "Environmentally adjusted $\delta^{13}$C thresholds for accurate detection of C$_4$ plant consumption in Europe" [Data set]. In Communications Earth & Environment. Zenodo. https://doi.org/10.5281/zenodo.17571650 [ref. 27 in this paper]. The data to reproduce the ecozone clusters is available from this open access repository: Kempf, M. (2025): Related files to: Depaermentier, MLC; Kempf, M; Motuzaitė Matuzevičiūtė, G: Environmentally adjusted $\delta^{13}$C thresholds for accurate detection of C$_4$ plant consumption in Europe (2025) [Data set]. Zenodo. https://doi.org/10.5281/zenodo.15695070 [ref. 147 in this paper]. Climate variables used in this article are freely available from Karger et al. (2017)[143]: https://chelsa-climate.org/ (last accessed 19th of June 2025). The Digital Elevation Model (DEM) can be downloaded from the USGS earthexplorer server: https://earthexplorer.usgs.gov/, last accessed 19th of June 2025.

## Code availability

The code to reproduce the ecozone clusters is available from this open access repository: Kempf, M. (2025): Related files to: Depaermentier, MLC; Kempf, M; Motuzaitė Matuzevičiūtė, G: Environmentally adjusted $\delta^{13}$C thresholds for accurate detection of C$_4$ plant consumption in Europe (2025) [Data set]. Zenodo. https://doi.org/10.5281/zenodo.15695070 [ref. 147 in this paper].

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

## Acknowledgements

M.L.C.D. and G.M.M. were funded by the European Union with a Consolidator Grant awarded to Giedrė Motuzaitė Matuzevičiūtė (ERC-CoG, MILWAYS, 101087964). Views and opinions expressed are those of the authors only and do not necessarily reflect those of the European Union or the European Research Council Executive Agency. Neither the European Union nor the granting authority can be held responsible for them. MK's research is funded by the Swiss National Science Foundation (SNSF/SNF): Project EXOCHAINS − Exploring Holocene Climate Change and Human Innovations across Eurasia (SNSF grant number: TMPFP2_217358).

## Author contributions

Conceptualization: M.L.C.D., M.K. and G.M.M. Isotope data collection and formal analyses: M.L.C.D. Environmental and cluster analyses: M.K. Writing: M.L.C.D and M.K. Editing: M.L.C.D., M.K., G.M.M. Visualisation: M.K. and M.L.C.D. Revision: M.L.C.D., M.K.

## Competing interests

The authors declare no competing interests.
