## [Transparent Peer Review file · Communications Earth & Environment]

Environmentally adjusted $\delta^{13}\text{C}$ thresholds for accurate detection of C4 plant consumption in Europe

Corresponding Author: Dr Margaux Depaermentier

This manuscript has been previously reviewed at another Nature Portfolio journal. This document only contains reviewer comments and rebuttal letters for versions considered at Communications Earth & Environment.

Version 0:

Decision Letter:

Dear Dr Depaermentier,

Please accept our sincere apologies for the unusually long delay in sending a decision on your manuscript.

Your manuscript titled "Environmentally adjusted threshold $\delta^{13}\text{C}$ values for C4 plant consumption in Europe" has now been seen by 3 reviewers, and we include their comments at the end of this message. They find your work of interest, but some important points are raised. We are interested in the possibility of publishing your study in Communications Earth & Environment, but would like to consider your responses to these concerns and assess a revised manuscript before we make a final decision on publication.

In particular, please ensure that the revised manuscript meets the following editorial threshold:

- Fully explain the limitations of your approach and the associated uncertainties, and account for their implications in your interpretations and conclusions.

We therefore invite you to revise and resubmit your manuscript, along with a point-by-point response that takes into account the points raised. Please highlight all changes in the manuscript text file.

Please submit your point-by-point responses as a separate file, distinct from your cover letter where you can add responses to the Editors' comments that you do not want to be made available to the reviewers. Word files are preferred. We recommend that any figures, tables or graphs that are included in the response to reviewers are also included in the main article or Supplementary Information.

Please use the following link to submit your revised manuscript, point-by-point response to the referees' comments (which should be in a separate document to any cover letter), a tracked-changes version of the manuscript (as a PDF file) and the completed checklist:

Link Redacted

We hope to receive your revised paper within six weeks; please let us know if you aren't able to submit it within this time so that we can discuss how best to proceed. If we don't hear from you, and the revision process takes significantly longer, we may close your file. In this event, we will still be happy to reconsider your paper at a later date, as long as nothing similar has been accepted for publication at Communications Earth & Environment or published elsewhere in the meantime.

Please do not hesitate to contact us if you have any questions or would like to discuss these revisions further. We look

forward to seeing the revised manuscript and thank you for the opportunity to review your work.

Best regards,

Nicola Colombo, PhD
Associate Editor, Communications Earth & Environment
Consulting Editor, Communications Sustainability

EDITORIAL POLICIES AND FORMATTING

We ask that you ensure your manuscript complies with our editorial policies. Please ensure that the following formatting requirements are met, and any checklist relevant to your research is completed and uploaded as a Related Manuscript file with the revised article.

- Behavioural and social science
- Ecological, evolutionary & environmental sciences
- Life sciences

Furthermore, please align your manuscript with our format requirements, which are summarized on the following checklist: <https://www.nature.com/documents/commsj-phys-style-formatting-checklist-article.pdf> Communications Earth & Environment formatting checklist

and also in our style and formatting guide <https://www.nature.com/documents/commsj-phys-style-formatting-guide-accept.pdf> Communications Earth & Environment formatting guide .

*** DATA: Communications Earth & Environment endorses the principles of the Enabling FAIR data project (<http://www.copdess.org/enabling-fair-data-project/>). We ask authors to make the data that support their conclusions available in permanent, publically accessible data repositories. (Please contact the editor if you are unable to make your data available).

All Communications Earth & Environment manuscripts must include a section titled "Data Availability" at the end of the Methods section or main text (if no Methods). More information on this policy, is available at <http://www.nature.com/authors/policies/data/data-availability-statements-data-citations.pdf>.

If a community resource is unavailable, data can be submitted to generalist repositories such as <https://figshare.com/> or <http://datadryad.org/> Dryad Digital Repository. Please provide a unique identifier for the data (for example a DOI or a permanent URL) in the data availability statement, if possible. If the repository does not provide identifiers, we encourage authors to supply the search terms that will return the data. For data that have been obtained from publically available sources, please provide a URL and the specific data product name in the data availability statement. Data with a DOI should be further cited in the methods reference section.

REVIEWER COMMENTS:

Reviewer #1 (Remarks to the Author):

This paper presents a compilation of a large dataset of stable carbon isotope values (d13C) of plants from archaeological and modern contexts across Europe. It provides an opportunity to assess variability in d13C values of plants grown across

distinct ecological zones in Europe. However, in my view, the paper does not offer as large a contribution to knowledge as the manuscript purports. The threshold that the authors say is commonly used to identify C4 consumption (-18‰) is not actually used as systematically in paleodietary studies as the authors say because of the uncertainties attached to it; so the gap that the paper is trying to fill (i.e., point out that mixed C3/C4 diets have different thresholds in different ecological zones) is exaggerated.

Additionally, I am worried about the data that went into the model. The authors state that they used plant values that have not been corrected for the charring effect. The reason that is given is “in order to enhance the comparability between datasets” (line 313). However, the charring offset is real (Charles et al. 2015; Nitch et al. 2015; Styring et al. 2024) and should not be ignored. What if grains/seeds at some sites were charred at higher temperatures and therefore their $\delta^{13}\text{C}$ values are offset by varying amounts? If this was the case, then this variability introduced by charring could obscure the relationship that the authors are trying to identify between $\delta^{13}\text{C}$ and latitude/ecological zones.

In addition to charring, I have other concerns about the data itself:

1. What if the crops represented in the dataset were cultivated under varying growing conditions, particularly soil moisture? The authors acknowledged growing conditions on line 106, but only with reference to samples from the UK. To properly identify a relationship between latitude/ecological zones and $\delta^{13}\text{C}$, it would be better to use measurements of wild plants rather than cultivated plants, since the values of cultivated plants can be anthropogenically manipulated.
2. What was the measurement error and how do you know that the different datasets (from the original source studies) are comparable? Szpak et al. (2017) discuss how important it is to consider both measurement precision, measurement accuracy, and an overall estimate of measurement error, especially for combining datasets in meta-analyses.
3. Lastly, the authors mention that some of the plant $\delta^{13}\text{C}$ measurements come from AMS radiocarbon laboratories, and it needs to be pointed out that $\delta^{13}\text{C}$ values that are done alongside radiocarbon measurements are often not calibrated the same way as $\delta^{13}\text{C}$ measurements done for Paleodietary studies (Vaiglova et al. 2023)

It is not entirely clear why the authors used r^2 values in their analyses. I would be more interested to see the Pearson's r value in order to assess the strength of correlation. More broadly though, it seems that there is some confusion between what is a small relationship and what is a weak relationship. A small relationship can be small in terms of its size (for example a decrease of -0.1‰ for every degree of latitude) but it can still be strong because the datapoints are close to the line of best fit and Pearson's r is closer to 1 than it is to 0. A relationship that is weak refers to cases where the datapoints are not close to the line and therefore appear as a shotgun blast (for example, what we see in Figure 3C with $\delta^{13}\text{C}$ vs. latitude). Throughout the paper, what we are looking at seem to be weak relationships, which makes it really hard to quantify their effects.

To improve this paper, I suggest the authors consider how best to present the results on the variability of plant isotope values. At the moment, there isn't one figure that really stands out and carries the main message of the paper. But more importantly, a better consideration of uncertainty is needed. For example, a statement like

“Our model shows that in the Baltic and Nordic countries, human or animal collagen $\delta^{13}\text{C}$ values around -19.8‰ already reflect a low C4 input within a primarily C3-based diet. In the Mediterranean, the same C4 input would result in collagen $\delta^{13}\text{C}$ values above -17.0 or -16.0‰ . (Figs. 5A-B and Tab. S2)” (line 216)

really need to be accompanied by error bars, because none of these modelled estimates are free from uncertainty.

Charles et al. 2015, “Nor ever lightning char thy grain”: establishing archaeologically relevant charring conditions and their effect on glume wheat grain morphology.

Nitch et al. 2015, Calculating a statistically robust $\delta^{13}\text{C}$ and $\delta^{15}\text{N}$ offset for charred cereal and pulse seeds.

Styring et al. 2024 Recommendations for stable isotope analysis of charred archaeological crop remains.

Szpak et al. 2017 Best practices for calibrating and reporting stable isotope measurements in archaeology.

Vaiglova et al. 2023 Best practices for selecting samples, analyzing data, and publishing results in isotope archaeology.

Reviewer #2 (Remarks to the Author):

This is a superbly written article to fill a gap in this topic. To be able to provide a new model with clear data demonstrating the variability between ecozones is both exciting and pertinent to so many sub-disciplines of archaeology. The questions posed in the introduction were addressed thoroughly, and supplemented with figures that illustrate the data well.

The statistical analyses are good, but the p-values and other results should be listed, either in text or in a supplemental file.

There are some minor comments on a few sentences, and you may not agree with them; my recommendations are for improved clarity as a first-time reader of this article. These are marked in the pdf document.

Reviewer #3 (Remarks to the Author):

General Evaluation

This manuscript addresses a central issue in bioarchaeology, paleoecology, and isotopic research: the inadequacy of using a single fixed $\delta^{13}\text{C}$ threshold (commonly -18‰ in collagen) to identify C4 plant consumption. The authors compiled an extensive dataset of over 3,500 $\delta^{13}\text{C}$ values from charred archaeological C3 and C4 cereals, covering more than 200 sites across Europe and the Mediterranean (8000 BCE–1800 CE). By applying a combination of statistical analyses (linear models, one-way ANOVA, ecozone-based clustering), they demonstrate that $\delta^{13}\text{C}$ values vary significantly depending on ecological and geographical contexts and propose an innovative ecozone-based model to define locally adapted thresholds for identifying C4 consumption.

Strengths

- The dataset is exceptionally large, well-curated, and based on both published sources and recently updated open-access databases (IsoArch, MAIA, Isotòpia, etc.). The explicit separation of C3 and C4 datasets, along with comparison to AMS-based values, enhances transparency.
- The introduction clearly identifies research questions, situates the study within both archaeological and ecological frameworks, and explains the biological basis of isotopic differences between C3 and C4 plants. I suggest the authors more explicitly highlight in the introduction that the ecozone-based approach represents a methodological innovation.
- The isotopic dataset section rigorously describes data acquisition and selection criteria; it would be useful to add the number of publications from which data were sourced.
- Statistical methods are described with great transparency regarding datasets, R packages, and code, ensuring reproducibility. This level of detail is commendable, although some information could be moved to Supplementary Materials to avoid overloading the main text.
- The Results section is well-organized into clear thematic subsections; figures and tables are appropriate and effectively support the text. Weak correlations are reported without overinterpretation.
- The Discussion effectively conveys the central finding: $\delta^{13}\text{C}$ thresholds for detecting C4 consumption cannot be fixed but must be context-specific. The ecozone-based model provides a valuable and innovative framework, integrating ecological, agricultural, and dietary perspectives.

Major Concerns

- The manuscript acknowledges geographic and chronological representation imbalances but does not fully assess their impact on the model. A more explicit discussion would strengthen the study.
- In some sections, causal interpretations (e.g., United Kingdom, Denmark) appear already in the Results, blurring description and interpretation.
- Theoretical diets (100% cereals or with 10–20% C4) are useful as baselines but are unrealistic. This limitation is acknowledged but should be emphasized more clearly.
- Using estimated values when local data are unavailable is understandable, but the additional uncertainty should be quantified or at least discussed more explicitly.

Minor Comments

- A more concise concluding paragraph in the Discussion summarizing advantages, limitations, and future perspectives would strengthen the final message.

In conclusion, this manuscript represents a significant and innovative contribution, with an ecozone-based model that improves the interpretation of isotopic data in the European context. Some methodological clarifications and greater emphasis on dataset limitations would enhance robustness, but overall it is a solid study of high relevance to bioarchaeology, ecology, and paleontology.

** Visit Nature Portfolio's author and referees' website at www.nature.com/authors for information about policies, services and author benefits**

Communications Earth & Environment is committed to improving transparency in authorship. As part of our efforts in this direction, we are now requesting that all authors identified as 'corresponding author' create and link their Open Researcher and Contributor Identifier (ORCID) with their account on the Manuscript Tracking System prior to acceptance. ORCID helps the scientific community achieve unambiguous attribution of all scholarly contributions. You can create and link your ORCID from the home page of the Manuscript Tracking System by clicking on 'Modify my Springer Nature account' and following the instructions in the link below. Please also inform all co-authors that they can add their ORCID to their accounts and that they must do so prior to acceptance.

Version 1:

Decision Letter:

Dear Dr. Depaermentier,

Your manuscript titled "Environmentally adjusted threshold $\delta^{13}C$ values for C4 plant consumption in Europe" has now been seen by our reviewers, whose comments appear below. In light of their advice we are delighted to say that we are happy, in principle, to publish a suitably revised version in Communications Earth & Environment.

We therefore invite you to revise your paper one last time to address the remaining concerns of our reviewers. At the same time we ask that you edit your manuscript to comply with our format requirements and to maximise the accessibility and therefore the impact of your work.

EDITORIAL REQUESTS:

*****Please take care to match our formatting and policy requirements. We will check revised manuscript and return manuscripts that do not comply. Such requests will lead to delays. *****

SUBMISSION INFORMATION:

OPEN ACCESS:

Communications Earth & Environment is a fully open access journal. Articles are made freely accessible on publication. For further information about article processing charges, open access funding, and advice and support from Nature Portfolio, please visit <https://www.nature.com/commsenv/open-access>

Link Redacted

Best regards,

Nicola Colombo, PhD
Associate Editor, Communications Earth & Environment
Consulting Editor, Communications Sustainability

REVIEWERS' COMMENTS:

Reviewer #1 (Remarks to the Author):

The authors have done an excellent job going through all the suggested comments and edits with careful attention to detail. All of my queries have been sufficiently addressed, and I am happy with their response to the reports from the other reviewers as well. I suggest that the paper is accepted and look forward to seeing it published. For the final proofs, please make sure to use a comma instead of an apostrophe for numbers in the thousands (e.g., 4,000 d13C values instead of 4'000 d13C values; line 22, Abstract).

Reviewer #3 (Remarks to the Author):

The authors have appropriately addressed the reviewers' comments, effectively clarifying the doubts raised. They have taken inspiration from the suggestions received and made substantial revisions to the article. In particular, the introduction of new statistical analyses (using Pearson correlation coefficient and standard deviation), the revision of the main figures, and the addition of supplementary tables presenting the test results have improved the robustness and transparency of the results. Furthermore, the expanded discussion has contributed to making the study overall more solid.

** Visit Nature Portfolio's author and referees' website at www.nature.com/authors for information about policies, services and author benefits**

Response to reviewers

Dear reviewers,

We would like to thank you very much for your insightful comments and suggestions. The constructive review helped us improve this paper and we really hope that our revision will meet your expectations.

Among the major changes, you will find a more detailed discussion on the limits of our approach with respect to the 100% grain-based diet used for the model, on the difficulty to assess whether the datasets are comparable with each other due to the diversity of analytical and calibration procedures, on the isotope variability within each presented geographical entity (from the site to the ecozone level), and on the various factors that may be responsible for such variability. In particular, we added the standard deviation (SD) values to each median value mentioned in the text (which may reflect this variability even better than the MAD that we were using so far, but which were not explicitly mentioned in the first version of the text) and we underlined this related variability and the importance to consider ranges instead of single threshold values even more than in the first version. We also implemented Pearson's correlations in addition to the linear models and ANOVA tests used so far and explicitly listed all results of the statistical tests in one new table (Tab. S3) as well as in the text when suitable. We furthermore revised two figures (Figs. 5 and 7) in order to focus on the most suitable model output (while we restricted the results of the second model to the supplementary files only) and used the gained space to better highlight the SD values directly on these figures (especially for Fig. 5, as it was already present in the previous version of Fig. 7).

To better exemplify the outcome of our paper, we clarified 1) our terminology by clearly defining what this paper offers: a European-wide $\delta^{13}\text{C}$ baseline for C3 grains and an environmentally adjusted threshold range of $\delta^{13}\text{C}$ values for C4 identification from the site to the ecozone level; and 2) our approach by clearly defining the pros and cons of the point-/site-based approach versus ecozone-based interpolation approach. The revised figures and the more detailed discussion now better support and transfer our message, while the newly created graphical abstract provides a clear overview of our workflow and outcome.

We would also like to inform you that we added some new relevant papers published in the meantime since our first submission and hence filled some geographical and chronological gaps. All tables, figures and values were updated accordingly. The fact that we revised all the numbers hitherto presented therefore does not mean that these needed to be revised, but only that new values were added and slightly modified each single value, without changing the overall outcome and results.

Please find a detailed point-by-point response to your reviews below (our responses in blue). We are looking forward to your decision on this revised version and thank you again very much for your recommendations and advice.

Kind regards,

Margaux L. C. Depaermentier, Michael Kempf and Giedre Motuzaitė Matuzevičiūtė

Point-by-point response to reviewers

REVIEWER COMMENTS:

Reviewer #1 (Remarks to the Author):

This paper presents a compilation of a large dataset of stable carbon isotope values ($\delta^{13}\text{C}$) of plants from archaeological and modern contexts across Europe. It provides an opportunity to assess variability in $\delta^{13}\text{C}$ values of plants grown across distinct ecological zones in Europe. However, in my view, the paper does not offer as large a contribution to knowledge as the manuscript purports. The threshold that the authors say is commonly used to identify C_4 consumption (-18‰) is not actually used as systematically in paleodietary studies as the authors say because of the uncertainties attached to it; so the gap that the paper is trying to fill (i.e., point out that mixed C_3/C_4 diets have different thresholds in different ecological zones) is exaggerated.

- There are indeed scholars who are dealing with the threshold value more cautiously and who are either already integrating the influence of other factors (environment, climate, analytical procedures, ...) or using statistical approaches to balance and adjust the threshold values for C_4 identification at the level of their study. But a huge number of papers use this threshold value throughout Europe, and we are convinced that there is an actual need to clarify this situation and make “users” of isotope data aware of this ecological variability. From the feedback that we received on this research at the 11th International Symposium on Biomolecular Archaeology, we can ensure that revising this threshold value and making aware of the environmentally triggered isotopic variability is actually needed. We therefore decided not to modify our introduction in accordance with this first comment.

Additionally, I am worried about the data that went into the model. The authors state that they used plant values that have not been corrected for the charring effect. The reason that is given is “in order to enhance the comparability between datasets” (line 313). However, the charring offset is real (Charles et al. 2015; Nitch et al. 2015; Styring et al. 2024) and should not be ignored. What if grains/seeds at some sites were charred at higher temperatures and therefore their $\delta^{13}\text{C}$ values are offset by varying amounts? If this was the case, then this variability introduced by charring could obscure the relationship that the authors are trying to identify between $\delta^{13}\text{C}$ and latitude/ecological zones.

- We agree with you that the charring effect is real and that it may have had an impact on at least some grains – if not on all – and it was difficult to take a decision in the first place. However, most studies reveal a particularly low (up to 0.11‰) and not systematic impact on grain $\delta^{13}\text{C}$, which implies that it does not make a dramatic difference to use corrected or uncorrected values when working on carbon. Even the offset of up to 0.2‰ with higher temperatures remain within the analytical error. To verify this assumption, we ran our analyses again in three different ways: 1) without any correction as suggested in the paper, 2) with a correction only for grain $\delta^{13}\text{C}$ values

that were also corrected by the authors of the original publication, and 3) with the same correction on all grain $\delta^{13}\text{C}$ values (using the -0.11‰ suggested by Nitsch et al. 20215 and used in the publications from this dataset that applied such correction). We are providing two examples of the minimal/insignificant impact of this correction on our results with the two figures below and suggest therefore keeping our strategy of using no correction on the data. We are, however, now providing more arguments for this decision in the main text (as explained earlier in this response).

In addition to charring, I have other concerns about the data itself:

1. What if the crops represented in the dataset were cultivated under varying growing conditions, particularly soil moisture? The authors acknowledged growing conditions on line 106, but only with reference to samples from the UK. To properly identify a relationship between latitude/ecological zones and $\delta^{13}\text{C}$, it would be better to use measurements of wild plants rather than cultivated plants, since the values of cultivated plants can be anthropogenically manipulated.

→ Thank you for raising this important issue. We have added a whole paragraph on this topic to the discussion to balance our interpretation and we also added arguments for justifying the use of crops for this study in the related methods section. In a nutshell, we agree and acknowledge that agricultural practices (as well as intra-species variability) probably have an impact on grain $\delta^{13}\text{C}$

values, and we promote a comparison with wild species to disentangle environmental and anthropogenic factors in, e.g., water availability. We nevertheless suggest that using crops is an important baseline when investigating humans, as this is an important part of human diet, regardless of agricultural practices, as this is how they eventually enter the human body. However, wild plants would be better for studying animal diet, yet these proxies are hardly/not available from archaeological context.

2. What was the measurement error and how do you know that the different datasets (from the original source studies) are comparable? Szpak et al. (2017) discuss how important it is to consider both measurement precision, measurement accuracy, and an overall estimate of measurement error, especially for combining datasets in meta-analyses.

→ Thank you so much for this important comment. Unfortunately, the information required to assess this (such as on calibration, accuracy or precision) was lacking in most of the papers. We have added the error values to each sample for which such a value was provided in the literature, and since it was usually very low, we hope that the datasets are overall comparable. We nevertheless added this aspect to the discussion to inform the reader about a potential impact of this issue on the isotopic variability presented in the paper – yet with the estimation that this impact was lower than those of the environmental variables.

3. Lastly, the authors mention that some of the plant $\delta^{13}\text{C}$ measurements come from AMS radiocarbon laboratories, and it needs to be pointed out that $\delta^{13}\text{C}$ values that are done alongside radiocarbon measurements are often not calibrated the same way as $\delta^{13}\text{C}$ measurements done for Paleodietary studies (Vaiglova et al. 2023)

→ There is unfortunately a misunderstanding. As stated in the introduction and method section, “This study draws on over 3’500 published $\delta^{13}\text{C}$ values from charred archaeological C3 and C4 grains **derived from Isotope-Ratio Mass Spectrometry (IRMS)**” / “**Only data obtained from Isotope-Ratio Mass Spectrometry (IRMS)** were selected for analyses.” and the AMS dataset was simply used as comparison dataset to check whether the same pattern would be observable despite the different calibration used to obtain these values. And as we concluded “AMS stable isotopic data lack precision and provide particularly wide and unusual $\delta^{13}\text{C}$ ranges for C4 grains. Therefore, **these data cannot be used to extend the IRMS dataset in this study.**”.

→ Therefore, this concern does not apply to our study. But to avoid this misunderstanding, we clarified in the method section even more why and in which context we used the AMS dataset.

It is not entirely clear why the authors used r^2 values in their analyses. I would be more interested to see the Pearson’s r value in order to assess the strength of correlation. More broadly though, it seems that there is some confusion between what is a small relationship and what is a weak relationship. A small relationship can be small in terms of its size (for example a decrease of -0.1‰ for every degree of latitude) but it can still be strong because the datapoints are close to the line of best fit and Pearson’s r is closer to 1 than it is to 0. A relationship that is weak refers to cases there the datapoints are not close to the line and therefore appear as a shotgun blast (for example, what we see in Figure 3C with $\delta^{13}\text{C}$ vs.

latitude). Throughout the paper, what we are looking at seem to be weak relationships, which makes it really hard to quantify their effects.

→ We have now applied Pearson's correlation as well, and we added the Pearson's r values in the text and in the new supplementary table (tab S3) dedicated to the results of statistical analyses. This approach confirms our results and shows them now more clearly. One more clarification at this point: the fact that many relationships showed with these tests are weak is no bad results for us, as the weak relationships related to chronology enabled us to decide to keep the dataset as a whole entity instead of splitting it into chronological bins. The patterns that we otherwise highlight in this paper show significant and moderate to strong relationships. We hope that the fact that all results are now summarised in table S3 will help clarify the strength and significance of the relationships tested.

To improve this paper, I suggest the authors consider how best to present the results on the variability of plant isotope values. At the moment, there isn't one figure that really stands out and carries the main message of the paper.

→ This is a good point. We have now decided to use and depict the SD values instead of the MAD values, which may better represent the variability of the data. We also focused the figures on the output of the model that we consider the best fit (instead of showing the other model as well in the figures, which may be confusing and misleading for the take-home message), and we could therefore use the gained space to better highlight the isotopic variability (especially in figure 5). We also created a graphical abstract to better summarise our approach and our outcome.

But more importantly, a better consideration of uncertainty is needed. For example, a statement like "Our model shows that in the Baltic and Nordic countries, human or animal collagen $\delta^{13}\text{C}$ values around -19.8‰ already reflect a low C4 input within a primarily C3-based diet. In the Mediterranean, the same C4 input would result in collagen $\delta^{13}\text{C}$ values above -17.0 or -16.0‰. (Figs. 5A-B and Tab. S2)" (line 216) really need to be accompanied by error bars, because none of these modelled estimates are free from uncertainty.

→ Thank you for your suggestion. We added the uncertainty range in the form of one standard deviation (SD) throughout the text as well as in the related tables to better underline this isotopic variability. We also added a more detailed discussion on this aspect. We also clarified in the method section how we present the results and why.

Charles et al. 2015, "Nor ever lightning char thy grain": establishing archaeologically relevant charring conditions and their effect on glume wheat grain morphology.

Nitch et al. 2015, Calculating a statistically robust $\delta^{13}\text{C}$ and $\delta^{15}\text{N}$ offset for charred cereal and pulse seeds.

Styring et al. 2024 Recommendations for stable isotope analysis of charred archaeological crop remains.

Szpak et al. 2017 Best practices for calibrating and reporting stable isotope measurements in archaeology.

Vaiglova et al. 2023 Best practices for selecting samples, analyzing data, and publishing results in isotope archaeology.

- Thank you for the clear literature recommendations. Except for the first one that is not on isotopes but only on morphology and archaeobotany, we have integrated these papers in the discussion of our article.

Reviewer #2 (Remarks to the Author):

This is a superbly written article to fill a gap in this topic. To be able to provide a new model with clear data demonstrating the variability between ecozones is both exciting and pertinent to so many sub-disciplines of archaeology. The questions posed in the introduction were addressed thoroughly, and supplemented with figures that illustrate the data well.

The statistical analyses are good, but the p-values and other results should be listed, either in text or in a supplemental file.

- Table S3 is now providing a summary of the p-values and related results/values of the ANOVA tests, linear models, and Pearson's correlation tests. These results are listed in the text as well in the results section – but were avoided in the discussion part as much as possible to minimise interruptions in the reading.

There are some minor comments on a few sentences, and you may not agree with them; my recommendations are for improved clarity as a first-time reader of this article. These are marked in the pdf document.

- Comment on the abstract: no action needed.
- Comment line 34: We added references to more general examples.
- Comment line 35: We accepted your suggestion and modified the sentence accordingly.
- Comment line 91: We accepted your suggestion and modified the sentence accordingly.
- Comment on caption of Figure 1: We accepted your suggestion and modified the sentence accordingly.
- Comment line 138: We accepted your suggestion and modified the sentence accordingly.
- Comment on Figure 3a: Yes, we have now ordered these by region to better exemplify the patterns.
- Comment line 190: We accepted your suggestion and modified the sentence accordingly.
- Comment line 197: We accepted your suggestion and modified the sentence accordingly.
- Comment lines 315 and 324: Yes, we have now added a table in which all results (p-values and other relevant information) are summarised for each statistical test conducted. This new table also includes reference to the figures that depict the tested comparison/variables.

Reviewer #3 (Remarks to the Author):

General Evaluation

This manuscript addresses a central issue in bioarchaeology, paleoecology, and isotopic research: the inadequacy of using a single fixed $\delta^{13}\text{C}$ threshold (commonly -18‰ in collagen) to identify C4 plant consumption. The authors compiled an extensive dataset of over 3,500 $\delta^{13}\text{C}$ values from charred archaeological C3 and C4 cereals, covering more than 200 sites across Europe and the Mediterranean (8000 BCE–1800 CE). By applying a combination of statistical analyses (linear models, one-way ANOVA, ecozone-based clustering), they demonstrate that $\delta^{13}\text{C}$ values vary significantly depending on ecological and geographical contexts and propose an innovative ecozone-based model to define locally adapted thresholds for identifying C4 consumption.

Strengths

- The dataset is exceptionally large, well-curated, and based on both published sources and recently updated open-access databases (IsoArch, MAIA, Isotòpia, etc.). The explicit separation of C3 and C4 datasets, along with comparison to AMS-based values, enhances transparency.
- The introduction clearly identifies research questions, situates the study within both archaeological and ecological frameworks, and explains the biological basis of isotopic differences between C3 and C4 plants. I suggest the authors more explicitly highlight in the introduction that the ecozone-based approach represents a methodological innovation.

→ Thank you. We added the adjectives “innovative and unprecedented” to introduce our ecozone-based approach in the introduction.

- The isotopic dataset section rigorously describes data acquisition and selection criteria; it would be useful to add the number of publications from which data were sourced.

→ Indeed. We added the number of publications used for this data collection and since we could add one more reference published in the meantime, enhancing our dataset in north-easter Europe, we also updated the n-number of C3 grains included and their related species.

- Statistical methods are described with great transparency regarding datasets, R packages, and code, ensuring reproducibility. This level of detail is commendable, although some information could be moved to Supplementary Materials to avoid overloading the main text.

→ Thank you. The methods and material section is already separated from the main text and following the journal’s guidelines, we are due to keep them as is and to not move parts of it to the supplementary materials.

- The Results section is well-organized into clear thematic subsections; figures and tables are appropriate

and effectively support the text. Weak correlations are reported without overinterpretation.

- The Discussion effectively conveys the central finding: $\delta^{13}\text{C}$ thresholds for detecting C4 consumption cannot be fixed but must be context-specific. The ecozone-based model provides a valuable and innovative framework, integrating ecological, agricultural, and dietary perspectives.

Major Concerns

- The manuscript acknowledges geographic and chronological representation imbalances but does not fully assess their impact on the model. A more explicit discussion would strengthen the study.

The new discussion now integrates much more aspects than the previous version, including a more detailed discussion on the chronological and geographical imbalance of the dataset – the geographical imbalance being discussed both in the middle of the discussion and again towards the end of the discussion section, when emphasizing the pros and cons of our point-based versus ecozone-based approaches.

- In some sections, causal interpretations (e.g., United Kingdom, Denmark) appear already in the Results, blurring description and interpretation.

→ Yes, we have modified this and kept the sample as a whole entity in the main figures but offered in the new table dedicated to the results of statistical tests (Tab S3) each time a version with all data and a version excluding UK and/or DK depending on what was relevant. This shall now enable to disentangle the impact (or lack of impact) of these outlier datasets on the whole isotopic dataset.

- Theoretical diets (100% cereals or with 10–20% C4) are useful as baselines but are unrealistic. This limitation is acknowledged but should be emphasized more clearly.

→ To underline the fact that this is a model based on an unrealistic gain-based diet, we emphasized this information even more in the main text and in the figure captions. We furthermore added a more detailed discussion of these limits in the corresponding paragraph.

- Using estimated values when local data are unavailable is understandable, but the additional uncertainty should be quantified or at least discussed more explicitly.

→ We added the uncertainty range in the form of one standard deviation (SD) throughout the text as well as in the related tables to better underline this isotopic variability. We also added a more detailed discussion on this aspect.

Minor Comments

- A more concise concluding paragraph in the Discussion summarizing advantages, limitations, and future perspectives would strengthen the final message.

→ We added a concluding paragraph to the discussion according to this suggestion. We also added a graphical abstract to emphasize the outcome of this paper.

In conclusion, this manuscript represents a significant and innovative contribution, with an ecozone-based model that improves the interpretation of isotopic data in the European context. Some methodological clarifications and greater emphasis on dataset limitations would enhance robustness, but overall it is a solid study of high relevance to bioarchaeology, ecology, and paleontology.

Response to reviewers

Dear reviewers,

We are very grateful for your positive feedback on the revised version of this paper.

As recommended by Reviewer #1, we replaced the apostrophe by a comma for numbers in the thousands – except for dates.

Kind regards,

Margaux L. C. Depaermentier, Michael Kempf and Giedre Motuzaitė Matuzeviciūtė

Environmentally adjusted threshold $\delta^{13}\text{C}$ values for C_4 plant consumption in Europe

Margaux L. C. Depaermentier^{1,*}, Michael Kempf^{2,3,*}, Giedrė Motuzaitė Matuzevičiūtė¹

¹ Faculty of History, Vilnius University, Vilnius, Lithuania.

² Department of Environmental Sciences, University of Basel, Basel, Switzerland.

³ Department of Geography, University of Cambridge, Cambridge, UK.

*Corresponding authors: margaux.depaermentier@if.vu.lt; michael.kempf@unibas.ch

[§]These authors contributed equally to this paper.

ORCID MLCD: 0000-0002-1801-3358; margaux.depaermentier@if.vu.lt

ORCID MK: 0000-0002-9474-4670; michael.kempf@unibas.ch

ORCID GMM: 0000-0001-9069-1551; giedre.motuzaitė@gmail.com

Abstract

Detecting C_4 plants consumption is central to investigating animal ecology, agricultural practices, dietary transitions, and socio-environmental adaptations. Carbon isotope analysis is used to detect C_4 plants in human or animal diets. However, the conventional $\delta^{13}\text{C}$ threshold values used to identify C_4 plant intake does not account for substantial ecological variability across Europe. By analyzing over 3'500 $\delta^{13}\text{C}$ values from European archaeological charred C_3 and C_4 grains, we establish adjusted $\delta^{13}\text{C}$ thresholds for C_4 consumption 
[revised manuscript text omitted]

$p=0.00723$; Fig. S3). When distinguishing between the geographical subsets UK ($R^2=0$,
$p=0.889$), Southern ($R^2=0.026$, $p=1.35\text{e-}13$), Central/Western ($R^2=0.191$, $p=7.25\text{e-}13$) and
Northern Europe (including Denmark: $R^2=0.066$, $p=1.8\text{e-}15$; excluding Denmark: $R^2=0.066$,
$p=1.8\text{e-}15$), the positive correlation between C₃ grain $\delta^{13}\text{C}$ values and the grain mean date is
particularly weak (Fig. S4A-E). In particular, the slightly stronger relationship observed for
Central/Western Europe (Fig. S4B) is biased by the youngest samples from Central France,
which exhibit particularly enriched $\delta^{13}\text{C}$ values (Fig. S4C). At the ecozone level, a weak but
significant increase in C₃ grains $\delta^{13}\text{C}$ values can be observed for ecozone 1 ($R^2=0.281$,
$p=9.31\text{e-}08$) and ecozone 17 only ($R^2=0.223$, $p=0.00157$) (Fig. 1, Fig. S5). The C₄ grains
dataset has a small sample size and covers a reduced geographical area, which does not
enable any diachronic analysis (Fig. S6A). The slight decrease in C₄ $\delta^{13}\text{C}$ values over time
therefore cannot be considered statistically significant (Fig. S6B). This implies that the C₃ and
the C₄ grains datasets were not subdivided into different chronological phases for the
subsequent analyses.

**Geographical isotope variability**

Splitting the C₃ grain dataset into geographical subsets (UK, Northern, Southern, and
Central/Western Europe) shows that the median $\delta^{13}\text{C}$ value of C₃ grains from Northern Europe
is approximately 1‰ lower than that from Southern Europe (Fig. 2A, Tab. 2). This confirms the
previous observations made on different types of samples such as faunal remains^{20,21} and
modern plants¹⁹. Despite the high latitude, however, C₃ grains from the UK exhibit among the
highest $\delta^{13}\text{C}$ values across time (Fig. 2A, Fig. S4), which can be related to the oceanic
climate^{20,21}. In Denmark, the low $\delta^{13}\text{C}$ values of the oldest half of the sample (c. 3700–3000
BCE) shift to particularly enriched $\delta^{13}\text{C}$ values for the most recent half of the sample (c. 1000
BCE–1000 CE) (Fig. S7). This might reflect changes in agricultural practices and soil
management following quality decrease starting from the Neolithic period^{28,29}.

**Fig. 2 | Geographical isotopic variability in C₃ grains. (A)** C₃ grains δ¹³C values in Europe. **(B)** C₃
 grain δ¹³C variability compared to latitudinal bins within continental Europe (excluding the UK). The
 middle line of the box represents the median value, the box is delimited by the quartiles Q1 on the left
 and Q3 on the right and contains the middle half of the sample, the horizontal lines completed by the
 outlier dots represent the extent of the data. The median and mean absolute deviation (MAD) values for
 each region and each latitudinal bin are listed in **Tab. 2**. A one-way ANOVA-test provided a p-value <2e-
 16 in both cases, with Df=3 for (A) and Df=6 for (B). A linear model ran between the latitude and the
 δ¹³C values shows a R² value of 0.159 and a p-value of 2.99e-12.

When excluding the UK from the dataset accordingly, we observe a significant decrease in C₃
 grain δ¹³C values with increasing latitude (Fig. 2B). Using the median values calculated for
 each latitudinal bin, the C₃ grains δ¹³C values from sites above 50° latitude are on average
 0.53 to 1.61‰ lower than those of grains from sites at latitudes below 50° (Fig. 2B, Tab. 2).
 This confirms the mean offset of around 1–2‰ between Southern and Northern Europe. In
 comparison, there is a mean variation of 0.08‰ among the median δ¹³C values of the
 latitudinal bins above 50° and approximately 0.37‰ among the median δ¹³C values of the bins
 below 50°. The difference between southern and northern sites is therefore substantial.

At the modern country level, the C₃ grains from Lithuania (n=153) are nearly 2.5‰ lower than
 those from Jordan (n=71) (Fig. 3A, Tab. 2) which exceeds the previously defined offset of 1–
 2‰ between Southern and Northern Europe^{19–21}. On the contrary, the Jordan sample is on
 average 0.8‰ more enriched than those from Greece (n=367) or Italy (n=476) despite their

shared southern location. Consequently, the North-South-dichotomy is not enough to
 characterize the different isotopic composition of grains from diverse parts of Europe and does
 not account for micro-regional environmental diversity.

**Fig. 3 | Differences in archaeological charred C₃ and C₄ grain $\delta^{13}\text{C}$ values in Europe. (A)** C₃ grain
 $\delta^{13}\text{C}$ values over the modern countries. **(B)** C₄ grain $\delta^{13}\text{C}$ values over modern countries. **(C)** The
 comparison between latitude and C₄ grain $\delta^{13}\text{C}$ values shows a weak but significant negative correlation.
 Boxplots are defined in Fig. 2. The median and MAD values for each modern country are listed in Tab.
 2. The red labels show non-representative sample sizes (n<10).

Similarly, the C₄ grain $\delta^{13}\text{C}$ values from Lithuania (mean: -10.88‰, n=20) are on average
 nearly 1‰ lower than those from Greece (mean: -10.19‰, n=12), France (mean: -10.15‰,
 n=16) or Poland (mean: -10.19‰, n=9,) (Fig. 3B-C; Tab. 2). The sample size from Spain (n=3,
 mean: -10.69‰) is too low to be considered representative. This trend confirms previous
 studies from China with depleted C₄ grain $\delta^{13}\text{C}$ values recorded at higher latitudes^{23,24}. The
 pattern is further supported by the larger C₄ grain $\delta^{13}\text{C}$ dataset resulting from Alpha Magnetic
 Spectrometer (AMS) in Europe³⁰, showing generally lower $\delta^{13}\text{C}$ values in Northern compared
 to Southern or Central Europe (Fig. S8). However, AMS stable isotopic data lack precision and
 provide particularly wide and unusual $\delta^{13}\text{C}$ ranges for C₄ grains. Therefore, these data cannot
 be used to extend the IRMS dataset in this study. Differences in local plant genotypes are
 considered more likely triggers for the isotopic variability than climate and water availability²².

Both C₃ and C₄ grains exhibit lower δ¹³C values in some northern regions relative to the rest
of Europe, leading to regional variation in the δ¹³C threshold for identifying C₄ consumption.

**Ecological isotopic variability**

The geographical isotopic variability is related to environmental factors captured in the
ecozone model. C₃ grain samples from Lithuania (n=153), Finland (n=35), and from parts of
Denmark (n=86) fall into the subhumid temperate lowlands of North-Eastern Europe
represented by ecozone 5 (n=274). Together with ecozone 17 (n=42) – represented by grains
from Norway only – these samples exhibit the lowest median δ¹³C values (-25.06 ± 1.03‰ and
188 -25.08 ± 1.48‰, respectively) for charred C₃ grains in Europe (Fig. 4, Tab. 2). The high
humidity, low temperature and low to moderately elevated topography of these ecozones can
explain the depleted δ¹³C values¹⁸. ~~On the contrary, the~~ ecozone 20 (n=474), represented by
balanced temperate plains scattered over Europe and including samples mostly from Denmark
and England, exhibit a much higher median δ¹³C value (-23.10 ± 0.93‰) and its range hardly
overlaps with the other northern samples. Beyond the influence of agricultural practices
mentioned above^{28,29}, this can be explained by higher temperatures and moderate humidity
characterizing this ecozone. In contrast, the 89 samples from ecozone 1 show the lowest
median δ¹³C value among the sites below 50° latitude (-24.13 ± 1.14‰). This reflects the mild
and moist conditions of this transition zone in ~~middle~~ altitude. In Southern Europe, the
ecozones 7 and 13, representing warm highlands in the Mediterranean area, show the highest
median δ¹³C values (-22.72 ± 0.93‰ and -22.47 ± 0.96‰, respectively), deriving from the drier
and warmer climatic conditions. The ecozones 2, 8 and 19 are scattered over wide areas of
Europe and are not related to extreme temperatures. Their isotopic ratios show intermediate
values. Ecozones 3, 12, 14 and 18 cannot be included in this isotope investigation due to the
small sample size.

**Fig. 4 | C₃ grains $\delta^{13}\text{C}$ values over the European ecozones clusters.** The ecozone numbers refer to
 the numbering in Tab. 1. Boxplots are defined in Fig. 2. The boxplots are ordered from top to bottom
 according to increasing temperature within the ecozone. The red labels underline non-representative
 sample sizes (n < 10). The median and MAD values for each ecozone can be found in Tab. 2.

Discussion

Building on the substantial geographical and ecological variation in isotope values within C₃
 (and to a lesser extent C₄) plants, it is essential to revise the commonly used $\delta^{13}\text{C}$ threshold
 for identifying C₄ consumption, such as -18.0‰ for mammal collagen (for example⁵). At each
 investigated site, the C₃ and C₄ grain $\delta^{13}\text{C}$ values from this dataset were used to create
 theoretical collagen $\delta^{13}\text{C}$ values for a diet based exclusively on these crops. This resulted in
 site-specific estimations for an overall C₃ grain-based diet with 10% to 20% C₄ grain inputs
 (Fig. 5 and Tab. S2). Our model shows that in the Baltic and Nordic countries, human or animal
 collagen $\delta^{13}\text{C}$ values around -19.8‰ already reflect a low C₄ input within a primarily C₃-based
 diet. In the Mediterranean, the same C₄ input would result in collagen $\delta^{13}\text{C}$ values above -17.0
 or -16.0‰. (Figs. 5A-B and Tab. S2).

It is generally accepted that at least 20% of dietary protein from an alternative source (such as
 C₄ compared to C₃ crops) is required to be detected in collagen³¹. Figures 5C–D illustrate
 corresponding theoretical $\delta^{13}\text{C}$ thresholds of approximately -18.0‰ in northern latitudes
 (except Denmark) and -16.0 to -15.0‰ at southern latitudes. However, this observation holds

only for diets based entirely on these crops. In practice, both humans and animals have more
 varied diets, and since collagen $\delta^{13}\text{C}$ values reflect the protein component of the diet¹⁵, plant
 intake contributes less to collagen isotopic composition than animal-derived proteins.

 **Fig. 5 | Modelled $\delta^{13}\text{C}$ values of the consumer's collagen tissues for a C_3 diet including 10% to**
 **20% C_4 inputs in a 100% grain-based-diet.** For both 10% (A–B) and 20% (C–D) C_4 inputs, results are
 presented first for all study sites (A and C), and subsequently for sites with sample sizes of $n \geq 10$ grains
 (B and D). The modelled collagen $\delta^{13}\text{C}$ mean, median, and MAD values for each site are listed in Tab.
 S2.

In regions where the diet includes significant proportions of mushrooms³², forest-derived foods
 influenced by the canopy effect^{33,34}, and/or freshwater fish^{35–37}, consumers are exposed to
 more depleted $\delta^{13}\text{C}$ values. These foods have a much stronger influence on collagen $\delta^{13}\text{C}$
 values than any enrichment from C_4 plants. Such conditions are highly plausible in the Baltics
 and in Scandinavia, where the threshold $\delta^{13}\text{C}$ value for C_4 consumption does not exceed the
 values suggested by the model with 10% C_4 input (Fig. 5A–B, Fig. 7). On the contrary, potential
 marine food consumption¹⁴ or sea-spray effects in coastal areas³⁸ need to be considered to
 avoid any over-estimation of C_4 plant intake. The combination of various isotope systems^{38–40}
 and/or the application of mixing models⁴¹ are powerful approaches to disentangle the diverse
 dietary sources.

The presented model based on 10% C₄ input is the most possible basis for future $\delta^{13}\text{C}$
 threshold value determination at the local or micro-regional scale (Fig. 5A-B, Tab. S2). The
 ecozone model can be used to create environmentally adjusted threshold $\delta^{13}\text{C}$ values for C₄
 consumption at the European scale (Fig. 6, Tab. S3). Our approach demonstrates that the
 threshold at -18.0‰ is mostly valid in the ecozones 1 and 2, whereas the northern ecozone 5
 and the widespread ecozone 17 require a threshold $\delta^{13}\text{C}$ value close to -19.0‰. In high-
 temperature Mediterranean lowlands (ecozone 16), European plains (ecozone 19), and
 Atlantic regions (ecozone 20), the threshold shifts to -17.0‰ (Fig. 6, Tab. S3). In the arid
 southern regions (ecozone 13, and partially ecozones 7 and 16), it approaches -16.0‰.

**Fig. 6 | Theoretical $\delta^{13}\text{C}$ threshold values for C₄ consumption across the European ecozone**
 **clusters.** The ecozone numbers refer to the numbering in Tab. 1. Boxplots are defined in Fig. 1. The
 boxplots are ordered from top to bottom according to increasing temperature within the ecozone. The
 red labels underline non-representative sample sizes ($n < 10$). The median and MAD values for each
 ecozone are listed in Tab. S3. The purple line represents the revised $\delta^{13}\text{C}$ threshold value for C₄
 consumption in mammal collagen (i.e., -18.0‰).

**Fig. 7 | Ecozone-based interpolation of the C₃ grain δ¹³C values and of the estimated threshold**
 **δ¹³C values for detecting C₄ consumers.** (A) Median δ¹³C (left) and MAD values (right) of charred C₃
 grains. (B) Median δ¹³C (left) and MAD (right) values of the estimated threshold for C₄ consumption.
 Both are interpolated at the ecozone-level using the 10% C₄ input model. Ecozones 3, 12, 14 and 18
 are left grey due to their too small sample sizes. Ecozones 4, 9, 10, 11 and 15 are left grey due to the
 absence of data. The ecozone's mean, median and MAD values for C₃ grains and for C₄ consumption
 are listed in Tab. 2 and Tab. S3, respectively. Figure © Michael Kempf.

Considerable isotopic variability is observed within each ecozone (Fig. 6, Tab. S3), indicating
 that a range, rather than a sharp threshold value, is more appropriate (Fig. 7, Tab. S3). To
 accurately estimate the actual local threshold values for C₄ consumption, it is essential to use
 δ¹³C and δ¹⁵N values from local and contemporaneous food resources, including crops (for
 example⁴²). In future bioarchaeological, ecological or palaeontological studies where local
 plants or food resources are unavailable for calculating an isotopic baseline, ecozone-based
 interpolations of C₃ grain δ¹³C values (Fig. 4, Fig. 7A–B, Tab. 2) and/or of threshold δ¹³C values
 (Fig. 6C–D, Fig. 7, Tab. S3) can be used to identify C₄ consumers in Europe.
[revised manuscript text omitted]

European background maps used to create figures 5 and 7 are vector map data from
<https://www.naturalearthdata.com/>, implemented using the package *rnaturalearth* in R-
Software^{110,121}.

Data availability

The isotopic dataset used in this study is available from the supplementary material of this
article. The data to reproduce the ecozone clusters are available from this repository: Kempf,
406 M. (2025): Related files to: Depaermentier, MLC; Kempf, M; Motuzaitė Matuzevičiūtė, G:
Environmentally adjusted threshold of $\delta^{13}\text{C}$ values for C_4 plant consumption in Europe (2025)
[Data set]. Zenodo. <https://doi.org/10.5281/zenodo.15695070>. Climate variables used in this
article are freely available from Karger et al. (2017)¹²⁰: <https://chelsea-climate.org/> (last
accessed 19th of June 2025). The Digital Elevation Model (DEM) can be downloaded from the
USGS earthexplorer server: <https://earthexplorer.usgs.gov/>, last accessed 19th of June 2025.
The code to reproduce the Ecozone model is available as a related file to this article or on
request from MK.

References

- 1. Sneha, M. L. & Arjun, R. Medicinal Knowledge In South India (During Neolithic to Early
Historic Period): An Analysis Of Staple Plant Dietary Nutrition. *CMDR Journal of Social*
*Research* **1**, 35–48 (2024).

- 2. Martin, L. *et al.* The place of millet in food globalization during Late Prehistory as
evidenced by new bioarchaeological data from the Caucasus. *Scientific Reports* **11**,
13124; 10.1038/s41598-021-92392-9 (2021).
- 3. Kaupová, S. *et al.* Dukes, elites, and commoners: dietary reconstruction of the early
medieval population of Bohemia (9th–11th Century AD, Czech Republic).
*Archaeological and Anthropological Sciences* **11**, 1887–1909; 10.1007/s12520-018-
0640-8 (2019).
- 4. Hakenbeck, S. E., Evans, J., Chapman, H. & Fothi, E. Practising pastoralism in an
agricultural environment: An isotopic analysis of the impact of the Hunnic incursions on
Pannonian populations. *PloS one* **12**, e0173079; 10.1371/journal.pone.0173079 (2017).
- 5. Lightfoot, E., Liu, X. & Jones, M. K. Why move starchy cereals? A review of the isotopic
evidence for prehistoric millet consumption across Eurasia. *World Archaeology* **45**, 574–
623; 10.1080/00438243.2013.852070 (2013).
- 6. Prasifka, J. & Heinz, K. The use of C3 and C4 plants to study natural enemy movement
and ecology, and its application to pest management. *International Journal of Pest*
*Management* **50**, 177–181; 10.1080/09670870410001731907 (2004).
- 7. Saarinen, J., Mantzouka, D. & Sakala, J. Aridity, Cooling, Open Vegetation, and the
Evolution of Plants and Animals During the Cenozoic. In *Nature through Time*, edited by
E. Martinetto, E. Tschopp & R. A. Gastaldo (Springer International Publishing, Cham,
2020), pp. 83–107.
- 8. Terry, R. C., Guerre, M. E. & Taylor, D. S. How specialized is a diet specialist? Niche
flexibility and local persistence through time of the Chisel-toothed kangaroo rat.
*Functional Ecology* **31**, 1921–1932; 10.1111/1365-2435.12892 (2017).
- 9. Drucker, D. G. *et al.* Ecology of large ungulates in the northeastern Iberian Peninsula
during the Upper Palaeolithic through stable isotopes and tooth wear analysis.
*Quaternary Environments and Humans* **2**, 100011; 10.1016/j.qeh.2024.100011 (2024).
- 10. Drucker, D. G. The Isotopic Ecology of the Mammoth Steppe. *Annu. Rev. Earth Planet.*
*Sci.* **50**, 395–418; 10.1146/annurev-earth-100821-081832 (2022).
- 11. O'Leary, M. H. Carbon isotope fractionation in plants. *Phytochemistry* **20**, 553–567;
10.1016/0031-9422(81)85134-5 (1981).
- 12. Farquhar, G. D. On the Nature of Carbon Isotope Discrimination in C4 Species.
*Functional Plant Biol.* **10**, 205; 10.1071/PP9830205 (1983).
- 13. Cerling, T. E., Wang, Y. & Quade, J. Expansion of C4 ecosystems as an indicator of
global ecological change in the late Miocene. *Nature* **361**, 344–345; 10.1038/361344a0
(1993).
- 14. Lee-Thorp, J. A. On isotopes and old bones. *Archaeometry* **50**, 925–950;
10.1111/j.1475-4754.2008.00441.x (2008).
- 15. Ambrose, S. H. Isotopic Analysis of Paleodiets: Methodological and Interpretative
Considerations. In *Investigations of ancient human tissue. Chemical analyses in*
*anthropology*, edited by M. K. Sandford (Gordon and Breach, Philadelphia, 1993), pp.
59–130.
- 16. Froehle, A. W., Kellner, C. M. & Schoeninger, M. J. Multivariate carbon and nitrogen
stable isotope model for the reconstruction of prehistoric human diet. *Am. J. Phys.*
*Anthropol.* **147**, 352–369; 10.1002/ajpa.21651 (2012).

- 17. Kellner, C. M. & Schoeninger, M. J. A simple carbon isotope model for reconstructing
prehistoric human diet. *Am. J. Phys. Anthropol.* **133**, 1112–1127; 10.1002/ajpa.20618
(2007).
- 18. van Klinken, G. J., Richards, M. P. & Hedges, R. E. M. An Overview of Causes for
Stable Isotopic Variations in Past European Human Populations. Environmental,
Ecophysiological, and Cultural Effects. In *Biogeochemical approaches to paleodietary*
*analysis. Advances in archaeological and museum science*, edited by S. H. Ambrose &
469 M. A. Katzenberg (New York, London, 2002), pp. 39–63.
- 19. Cooper, C. G., Cooper, M. D., Richards, M. P. & Schmitt, J. Geographic and seasonal
variation in $\delta^{13}\text{C}$ values of C3 plant arabidopsis: Archaeological implications. *Journal of*
*Archaeological Science* **149**, 105709; 10.1016/j.jas.2022.105709 (2023).
- 20. Hedges, R. E., Stevens, R. E. & Richards, M. Bone as a stable isotope archive for local
climatic information. *Quaternary Science Reviews* **23**, 959–965;
10.1016/j.quascirev.2003.06.022 (2004).
- 21. van Klinken, G. J., van der Plicht, J. & Hedges, R. E. M. Bone $^{13}\text{C}/^{12}\text{C}$ ratios reflect
(palaeo) climatic variations. *Geophysical Research Letters* **21**, 445–448;
10.1029/94GL00177 (1994).
- 22. Lightfoot, E. *et al.* Carbon and nitrogen isotopic variability in foxtail millet (*Setaria italica*)
with watering regime. *Rapid communications in mass spectrometry : RCM* **34**, e8615;
10.1002/rcm.8615 (2020).
- 23. Dong, Y. *et al.* The potential of stable carbon and nitrogen isotope analysis of foxtail and
broomcorn millets for investigating ancient farming systems. *Frontiers in plant science*
**13**, 1018312; 10.3389/fpls.2022.1018312 (2022).
- 24. An, C.-B. *et al.* Variability of the stable carbon isotope ratio in modern and
archaeological millets: evidence from northern China. *Journal of Archaeological Science*
**53**, 316–322; 10.1016/j.jas.2014.11.001 (2015).
- 25. Olson, D. M. *et al.* Terrestrial Ecoregions of the World: A New Map of Life on Earth.
*BioScience* **51**, 933; 10.1641/0006-3568(2001)051[0933:TEOTWA]2.0.CO;2 (2001).
- 26. Larsson, M., Bergman, J. & Olsson, P. A. Soil, fertilizer and plant density: Exploring the
influence of environmental factors to stable nitrogen and carbon isotope composition in
cereal grain. *Journal of Archaeological Science* **163**, 105935; 10.1016/j.jas.2024.105935
(2024).
- 27. Lightfoot, E. & Stevens, R. E. Stable isotope investigations of charred barley (*Hordeum*
*vulgare*) and wheat (*Triticum spelta*) grains from Danebury Hillfort: implications for
palaeodietary reconstructions. *Journal of Archaeological Science* **39**, 656–662;
10.1016/j.jas.2011.10.026 (2012).
- 28. Gron, K. J. *et al.* Archaeological cereals as an isotope record of long-term soil health
and anthropogenic amendment in southern Scandinavia. *Quaternary Science Reviews*
**253**, 106762; 10.1016/j.quascirev.2020.106762 (2021).
- 29. Hald, M. M. *et al.* Farming during turbulent times: Agriculture, food crops, and manuring
practices in Bronze Age to Viking Age Denmark. *Journal of Archaeological Science:*
*Reports* **58**, 104736; 10.1016/j.jasrep.2024.104736 (2024).
- 30. Filipović, D. *et al.* New AMS ^{14}C dates track the arrival and spread of broomcorn millet
cultivation and agricultural change in prehistoric Europe. *Sci Rep* **10**, 13698;
10.1038/s41598-020-70495-z (2020).

- 31. Hedges, R. E. M. Isotopes and red herrings: comments on Milner et al . and Lidén et al.
*Antiquity* **78**, 34–37; 10.1017/S0003598X00092905 (2004).
- 32. O'Regan, H. J., Lamb, A. L. & Wilkinson, D. M. The missing mushrooms: Searching for
fungi in ancient human dietary analysis. *Journal of Archaeological Science* **75**, 139–143;
10.1016/j.jas.2016.09.009 (2016).
- 33. Bonafini, M., Pellegrini, M., Ditchfield, P. & Pollard, A. M. Investigation of the 'canopy
effect' in the isotope ecology of temperate woodlands. *Journal of Archaeological
Science* **40**, 3926–3935; 10.1016/j.jas.2013.03.028 (2013).
- 34. Drucker, D. G., Bridault, A., Hobson, K. A., Szuma, E. & Bocherens, H. Can carbon-13
in large herbivores reflect the canopy effect in temperate and boreal ecosystems?
Evidence from modern and ancient ungulates. *Palaeogeography, Palaeoclimatology,
Palaeoecology* **266**, 69–82; 10.1016/j.palaeo.2008.03.020 (2008).
- 35. Robson, H. K. *et al.* Carbon and nitrogen stable isotope values in freshwater, brackish
and marine fish bone collagen from Mesolithic and Neolithic sites in central and northern
Europe. *Environmental Archaeology* **21**, 105–118; 10.1179/1749631415Y.0000000014
(2016).
- 36. Guiry, E. Complexities of Stable Carbon and Nitrogen Isotope Biogeochemistry in
Ancient Freshwater Ecosystems: Implications for the Study of Past Subsistence and
Environmental Change. *Front. Ecol. Evol.* **7**; 10.3389/fevo.2019.00313 (2019).
- 37. Häberle, S. *et al.* Carbon and nitrogen isotopic ratios in archaeological and modern
Swiss fish as possible markers for diachronic anthropogenic activity in freshwater
ecosystems. *Journal of Archaeological Science: Reports* **10**, 411–423;
10.1016/j.jasrep.2016.10.012 (2016).
- 38. Göhring, A., Hölzl, S., Mayr, C. & Strauss, H. Identification and quantification of the sea
spray effect on isotopic systems in α -cellulose ($\delta^{13}\text{C}$, $\delta^{18}\text{O}$), total sulfur ($\delta^{34}\text{S}$), and

[revised manuscript text omitted]

*Anthropol Sci* **12**; 10.1007/s12520-020-01210-2 (2020).
- 80. Gavériaux, F., Motta, L., Bailey, P., Brillì, M. & Sadori, L. Crop Husbandry at Gabii
During the Iron Age and Archaic Period: The Archaeobotanical and Stable Isotope
Evidence. *Environmental Archaeology* **29**, 370–383; 10.1080/14614103.2022.2101281
(2024).
- 81. Gavériaux, F. *et al.* L'alimentation des premières sociétés agropastorales du Sud de la
France: premières données isotopiques sur des graines et fruits carbonisés néolithiques
et essais de modélisation. *Comptes Rendus. Palevol.* (2021).
- 82. García-Collado, M. I. *et al.* First Direct Evidence of Agrarian Practices in the Alava
Plateau (Northern Iberia) During the Middle Ages Through Carbon and Nitrogen Stable
Isotope Analyses of Charred Seeds. *Environmental Archaeology*, 1–11;
10.1080/14614103.2022.2091725 (2022).
- 83. Fiorentino, G., Caracuta, V., Casiello, G., Longobardi, F. & Sacco, A. Studying ancient
crop provenance: implications from $\delta^{13}C$ and $\delta^{15}N$ values of charred barley in a
Middle Bronze Age silo at Ebla(NW Syria). *Rapid communications in mass spectrometry*
*: RCM* **26**, 327–335; 10.1002/rcm.5323 (2012).
- 84. Fiorentino, G. *et al.* Third millennium B.C. climate change in Syria highlighted by Carbon
stable isotope analysis of ^{14}C -AMS dated plant remains from Ebla. *Palaeogeography,*
*Palaeoclimatology, Palaeoecology* **266**, 51–58; 10.1016/j.palaeo.2008.03.034 (2008).

- 85. Fernández-Crespo, T., Ordoño, J., Bogaard, A., Llanos, A. & Schulting, R. A snapshot of
subsistence in Iron Age Iberia: The case of La Hoya village. *Journal of Archaeological*
*Science: Reports* **28**, 102037; 10.1016/j.jasrep.2019.102037 (2019).
- 86. Eklund, M. *Changing Agriculture. Stable isotope analysis of charred cereals from Iron*
*Age Öland*. (Master thesis, Stockholm University, Stockholm, 2019).
- 87. DiBenedetto, K. E. *Investigating Land Use by the Inhabitants of Western Cyprus During*
*the Early Neolithic* (2018).
- 88. Cortese, F. *et al. Isotopic reconstruction of the subsistence strategy for a Central Italian*
*Bronze Age community (Pastena cave, 2 nd millennium BCE)* (2022).
- 89. Bogaard, A. *et al. Crop manuring and intensive land management by Europe's first*
*farmers. Proceedings of the National Academy of Sciences of the United States of*
*America* **110**, 12589–12594; 10.1073/pnas.1305918110 (2013).
- 90. Bogaard, A. *et al. From Traditional Farming in Morocco to Early Urban Agroecology in*
*Northern Mesopotamia: Combining Present-day Arable Weed Surveys and Crop Isotope*
*Analysis to Reconstruct Past Agrosystems in (Semi-)arid Regions. Environmental*
*Archaeology* **23**, 303–322; 10.1080/14614103.2016.1261217 (2018).
- 91. Bernardini, S. *et al. New multi-proxy isotopic data on the Copper age of eastern Liguria.*
*Rivista di Scienze Preistoriche* **LXXIII S3**, 1037–1043 (2023).
- 92. Ben Makhad, S. *et al. Crop manuring on the Beauce plateau (France) during the second*
*iron age. Journal of Archaeological Science: Reports* **43**, 103463;
10.1016/j.jasrep.2022.103463 (2022).
- 93. Arous, J. L. *et al. Isotope and morphometrical evidence reveals the technological*
*package associated with agriculture adoption in western Europe. PNAS* **121**,
e2401065121; 10.1073/pnas.2401065121 (2024).
- 94. Arous, J. L. *et al. Identification of Ancient Irrigation Practices based on the Carbon*
*Isotope Discrimination of Plant Seeds: a Case Study from the South-East Iberian*
*Peninsula. Journal of Archaeological Science* **24**, 729–740; 10.1006/jasc.1997.0154
(1997).
- 95. Arous, J. L. *et al. Changes in carbon isotope discrimination in grain cereals from*
*different regions of the western Mediterranean Basin during the past seven millennia.*
*Palaeoenvironmental evidence of a differential change in aridity during the late*
*Holocene. Global Change Biology* **3**, 107–118; 10.1046/j.1365-2486.1997.00056.x
(1997).
- 96. Arous, J. L. & Buxó, R. Changes in Carbon Isotope Discrimination in Grain Cereals
From the North-Western Mediterranean Basin During the Past Seven Millenia.
*Functional Plant Biol.* **20**, 117; 10.1071/PP9930117 (1993).
- 97. Antanaitis, I. & Ogrinc, N. Chemical analysis of bone: stable isotope evidence of the diet
of Neolithic and Bronze Age people In Lithuania. *Istorija* **XLV**, 3–12 (2000).
- 98. Alagich, R., Gardeisen, A., Alonso, N., Rovira, N. & Bogaard, A. Using stable isotopes
and functional weed ecology to explore social differences in early urban contexts: The
case of Lattara in mediterranean France. *Journal of Archaeological Science* **93**, 135–
149; 10.1016/j.jas.2018.03.006 (2018).
- 99. Aguilera, M., Zech-Matterne, V., Lepetz, S. & Balasse, M. Crop Fertility Conditions in
North-Eastern Gaul During the La Tène and Roman Periods: A Combined Stable
Isotope Analysis of Archaeobotanical and Archaeozoological Remains. *Environmental*
*Archaeology* **23**, 323–337; 10.1080/14614103.2017.1291563 (2018).

- 100. Salesse, K. *et al.* IsoArch.eu: An open-access and collaborative isotope database for
bioarchaeological samples from the Graeco-Roman world and its margins. *Journal of*
*Archaeological Science: Reports* **19**, 1050–1055; 10.1016/j.jasrep.2017.07.030 (2018).
- 101. Martina Farese. MAIA: Mediterranean Archive of Isotopic dAta, 2023.
- 102. Giulia Formichella, Silvia Soncin and Carlo Coccozza. Isotòpia: A Stable Isotope
Database for Classical Antiquity, 2023.
- 103. Mantile, N., Fernandes, R., Lubritto, C. & Coccozza, C. IsoMedIta: A Stable Isotope
Database for Medieval Italy. *Res. Data J Humanit. Soc. Sci.* **8**, 1–13;
10.1163/24523666-bja10032 (2023).
- 104. Coccozza, C., Cirelli, E., Groß, M., Teegen, W.-R. & Fernandes, R. Presenting the
Compendium Isotoporum Medii Aevi, a Multi-Isotope Database for Medieval Europe.
*Scientific data* **9**, 354; 10.1038/s41597-022-01462-8 (2022).
- 105. Graven, H., Keeling, R. F. & Rogelj, J. Changes to Carbon Isotopes in Atmospheric CO₂
Over the Industrial Era and Into the Future. *Global biogeochemical cycles* **34**,
e2019GB006170; 10.1029/2019GB006170 (2020).
- 106. Stroud, E., Charles, M., Bogaard, A. & Hamerow, H. Turning up the heat: Assessing the
impact of charring regime on the morphology and stable isotopic values of cereal grains.
*Journal of Archaeological Science* **153**, 105754; 10.1016/j.jas.2023.105754 (2023).
- 107. Nitsch, E. K., Charles, M. & Bogaard, A. Calculating a statistically robust $\delta^{13}\text{C}$ and $\delta^{15}\text{N}$
offset for charred cereal and pulse seeds. *STAR: Science & Technology of*
*Archaeological Research* **1**, 1–8; 10.1179/2054892315Y.0000000001 (2015).
- 108. Varalli, A., D'Agostini, F., Madella, M., Fiorentino, G. & Lancelotti, C. Charring effects on
stable carbon and nitrogen isotope values on C₄ plants: Inferences for archaeological
investigations. *Journal of Archaeological Science* **156**, 105821;
10.1016/j.jas.2023.105821 (2023).
- 109. Teira-Brión, A., Stroud, E., Charles, M. & Bogaard, A. The effects of charring on
morphology and stable carbon and nitrogen isotope values of common and foxtail millet
grains. *Front. Environ. Archaeol.* **3**; 10.3389/fearc.2024.1473593 (2024).
- 110. R Core Team. *R: A language and environment for statistical* (R Foundation for Statistical
Computing, Vienna, 2021).
- 111. Bates, D., Mächler, M., Bolker, B. & Walker, S. Fitting Linear Mixed-Effects Models
Using lme4. *J. Stat. Soft.* **67**; 10.18637/jss.v067.i01 (2015).
- 112. Hijmans, R. J. *_terra: Spatial Data Analysis_* (2024).
- 113. Pebesma, E. Simple Features for R: Standardized Support for Spatial Vector Data. *The*
*R Journal* **10**, 439; 10.32614/RJ-2018-009 (2018).
- 114. Pebesma, E. & Bivand, R. *Spatial Data Science* (Chapman and Hall/CRC, New York,
2023).
- 115. Warnes, G. *et al.* *_gtools: Various R Programming Tools_* (2023).
- 116. Wickham, H., François, R., Henry, L., Müller, K. & Vaughan, D. *_dplyr: A Grammar of*
*Data Manipulation_* (2023).
- 117. Wickham, H. *Ggplot2. Elegant graphics for data analysis* (Springer Science+Business
Media, LLC, New York, NY, 2016).
- 118. Dunnington, D. *_ggspatial: Spatial Data Framework for ggplot2_* (2023).
- 119. Auguie, B. *_gridExtra: Miscellaneous Functions for "Grid" Graphics* (2017).

120. Karger, D. N. *et al.* Climatologies at high resolution for the earth's land surface areas.
*Scientific data* **4**, 170122; 10.1038/sdata.2017.122 (2017).

121. South, A., Michael, S. & Massicotte, P. *rnatuarearthdata: World Vector Map Data from*
*Natural Earth Used in 'rnatuarearth'* (2017).

Acknowledgements

MLCD and GMM were funded by the European Union with a Consolidator Grant awarded to
Giedrė Motuzaitė Matuzevičiūtė (ERC-CoG, MILWAYS, 101087964). Views and opinions
expressed are those of the authors only and do not necessarily reflect those of the European
Union or the European Research Council Executive Agency. Neither the European Union nor
the granting authority can be held responsible for them. MK's research is funded by the Swiss
National Science Foundation (SNSF/SNF): Project EXOCHAINS - Exploring Holocene Climate
Change and Human Innovations across Eurasia (SNSF grant number: TMPFP2_217358).

Author contributions

Conceptualization: MLCD, MK, and GMM.

Data collection: MLCD.

Formal analyses: MLCD.

Environmental analyses: MK.

Writing: MLCD, MK.

Editing: MK, GMM.

Visualisation: MK, MLCD.

Competing interests

The authors declare no conflict of interest.

Materials & Correspondence

The corresponding author is MLCD. The isotopic dataset used in this study is available from
the supplementary material of this article. The data to reproduce the ecozone clusters are
available from this repository: Kempf, M. (2025): Related files to: Depaermentier, MLC; Kempf,
808 M; Motuzaitė Matuzevičiūtė, G: Environmentally adjusted threshold of $\delta^{13}\text{C}$ values for C_4 plant
consumption in Europe (2025) [Data set]. Zenodo. <https://doi.org/10.5281/zenodo.15695070>.
Climate variables used in this article are freely available from Karger et al. (2017):
<https://chelsa-climate.org/> (last accessed 19th of June 2025). The Digital Elevation Model
(DEM) can be downloaded from the USGS earthexplorer server:
<https://earthexplorer.usgs.gov/>, last accessed 19th of June 2025. The code to reproduce the
Ecozone model is available as a related file to this article or on request from MK.

Tables

cluster	mean TMP	mean CWB	mean DEM	SD TMP	SD CBW	SD DEM	Description (TMP CWB DEM)
---------	----------	----------	----------	--------	--------	--------	-------------------------------

0.429	0.405	0.205	0.039	0.017	0.021	Mild Moderate Middle
0.362	0.343	0.06	0.019	0.01	0.013	Cool Moderately Dry Low
0.111	1	0.343	0.053	0.05	0.056	Very Cold Extremely Humid High
0.116	0.429	0.065	0.02	0.007	0.015	Cold Humid Low
0.242	0.392	0.051	0.02	0.009	0.011	Cold Moderate Low
NA	NA	NA	NA	NA	NA	NA [unclassified waterbody]
0.659	0.248	0.296	0.04	0.019	0.024	Hot Dry High
0.398	0.544	0.083	0.05	0.019	0.026	Mild Very Humid Moderately Low
0.115	0.51	0.296	0.051	0.022	0.03	Cold Humid High
0.938	0.012	0.269	0.039	0.017	0.027	Very Hot Very Dry Middle
0	0.521	1	0.065	0.051	0.058	Very Cold Very Humid Alpine
0.388	0.406	0.43	0.043	0.024	0.03	Mild Humid Very High
0.633	0.184	0.479	0.053	0.026	0.03	Hot Very Dry Very High
0.329	0.318	0.718	0.051	0.025	0.043	Cool Moderately Dry Alpine
0.091	0.584	0.579	0.07	0.035	0.042	Very Cold Very Humid Very High
0.799	0.226	0.092	0.032	0.021	0.027	Hot Dry Moderately Low
0.297	0.749	0.131	0.052	0.031	0.038	Cool Extremely Humid Middle
1	0	0.078	0.031	0.018	0.023	Very Hot Very Dry Moderately Low
0.53	0.306	0.045	0.034	0.012	0.015	Warm Dry Very Low
0.473	0.395	0.046	0.032	0.011	0.015	Warm Moderate Very Low

Tab. 1 | K-means cluster ecozone cluster summary table including description. TMP: temperature; CWB: moisture availability; DEM: digital elevation model (i.e., topography); SD: standard deviation. Cluster 6 represents unclassified water bodies (NA; -99999).

Variable	n C ₃ grains	Median C ₃ grain $\delta^{13}\text{C}$ (‰)	MAD for C ₃ grain $\delta^{13}\text{C}$ (‰)	Comment (C ₃ grains)	n C ₄ grains	Median C ₄ grain $\delta^{13}\text{C}$ (‰)	MAD for C ₄ grain $\delta^{13}\text{C}$ (‰)	Comment (C ₄ grains)	
Latitude bins (in °)	[30-35[123	-22.79	0.76		0			
	[35-40[744	-23.12	1.02		0			
	[40-45[1187	-23.49	0.76		18	-10.49	0.44	
	[45-50[199	-23.40	1.04		13	-10.00	0.30	
	[50-55[275	-24.40	1.35		18	-10.345	0.28	
	[55-60[613	-24.02	1.41		11	-10.93	0.52	(ends at 55.69°)
	[60-63[82	-24.52	1.16		0			

Region	North	924	-24.20	1.38		20	-10.83	0.48	
	UK	249	-22.95	0.93		0			
	Central/ West	245	-23.49	1.05		22	-10.19	0.29	
	South	2054	-23.30	0.89		18	-10.49	0.44	
Modern country	Lebanon	2	-25.45	2.45	small sample size	0			
	Lithuania	153	-25.18	1.01		20	-10.83	0.48	
	Norway	63	-24.85	1.33		0			
	Finland	35	-24.42	0.92		0			
	Germany	50	-24.35	1.04		0			
	Hungary	4	-24.25	0.15	small sample size	0			
	Sweden	247	-24.20	1.04		0			
	Switzerland	7	-24.10	0.30	small sample size	0			
	Poland	52	-23.90	1.12		9	-10.19	0.15	small sample size
	Denmark	403	-23.70	1.51		0			
	Slovakia	10	-23.50	0.44		0			
	Bulgaria	22	-23.50	0.44		0			
	Greece	367	-23.50	0.76		12	-10.19	0.15	
	Italy	476	-23.50	0.89		0			
	Spain	654	-23.35	0.90		3	-10.69	0.15	small sample size
	France	206	-23.20	0.82		16	-10.15	0.36	
	Turkey	168	-23.12	0.79		0			
	Cyprus	8	-23.09	0.76	small sample size	0			
	Syria	216	-22.99	0.76		0			
England	249	-22.95	0.93		0				
Jordan	71	-22.72	1.01		0				

	Andorra	3	-22.10	0.37	small sample size	0			
	Georgia	6	-21.30	0.67	small sample size	0			
Ecozone cluster	1	89	-24.13	1.14		8	-10.44	0.37	small sample size
	2	440	-24.00	1.19		9	-10.19	0.15	small sample size
	3	4	-24.69	1.25	small sample size	0			
	5	274	-25.06	1.03		20	-10.83	0.48	
	7	149	-22.72	0.93		0			
	8	29	-23.76	0.83		0			
	12	3	-22.10	0.37	small sample size	0			
	13	37	-22.47	0.96		0			
	14	6	-21.30	0.67	small sample size	0			
	16	991	-23.20	0.90		12	-10.19	0.15	
	17	42	-25.08	1.48		0			
	18	3	-22.10	0.15	small sample size	0			
	19	931	-23.50	0.74		11	-10.10	0.30	
20	474	-23.10	0.93		0				

**Tab. 2 | Statistical summary for the C₃ and C₄ grain $\delta^{13}\text{C}$ values over the latitude bins, regions,**
**modern countries and ecozones.** The values indicated for small sample sizes are considered non-
representative for the related category.

0 500 1000 km

(c) Michael Kempf 2025
 Proj EPSG:3857; Grid EPSG:4326

Europe Ecozone Cluster

Very Cold Very Humid Very high	Cool Extremely Humid Middle	Cool Moderately Dry Low	Warm Moderate Very Low	Hot Dry Moderately Low
Very Cold Extremely Humid High	Cold Humid Low	Mild Very Humid Moderately Low	Warm Dry Very Low	Very Hot Very Dry Moderately Low
Very Cold Very Humid Alpine	Cold Moderate Low	Mild Humid Very High	Hot Very Dry Very High	Very Hot Very Dry Middle
Cold Humid High	Cool Moderately Dry Alpine	Mild Moderate Middle	Hot Dry High	Unclassified [NA]

A)**B)**
A) Regions ■ South ■ Central/West ■ North ■ UK

B)

C)